# COMPASS subunit Bre2 regulates chromatin remodeler Arp9 to control *Aspergillus flavus* aflatoxin synthesis and virulence

Zhenhong Zhuang[1], Minghui Sun[2], Dandan Wu[1], Dongmei Ma[2], Lin Chen[2], Xiaohua Pan[1], Hong Lin[1], Yu Li[1], Xuezhen Ma[1] & Shihua Wang [1] ✉

*Aspergillus flavus*, along with its notorious secondary metabolite aflatoxin B1 (AFB1), seriously endangers human health. Histone methyltransferase complex COMPASS (complex of proteins associated with Set1) plays a crucial role in regulating aflatoxin biosynthesis and virulence of *A. flavus*, but the underlying mechanism is unclear. Here, we find that Bre2, the key subunit of COMPASS, regulates AFB1 biosynthesis, fungal morphogenesis, and virulence through modulation of H3K4 methylation. ChIP-seq and biochemical analyses reveal that chromatin remodeling factor (CRF) Arp9 is directly targeted by Bre2, and Arp9 exerts bio-functions through interacting with the other CRFs such as RSC8, Arp7, and Sth1. ATAC-seq results indicate that Arp9 contributes to fungal pathogenicity by modulating chromatin conformation of genes that are involved in secondary metabolism, morphogenesis, and virulence. The study reveals an epigenetic signaling pathway mediated by chromatin remodeler Arp9 and provides a potential strategy for the control of pathogenic fungi and mycotoxins.

*A spergillus flavus* isa notorious fungus that produces toxic aflatoxins (AFs). It poses a serious threat to human health by food contamination[1–4]. Characterized by hepatotoxicity, teratogenicity, and immunotoxicity, AFs can be taken in through the respiratory, mucous, or even skin pathways. AFB1 is the most toxic and productive aflatoxin. It acts synergistically with hepatitis B virus (HBV), leading to significantly elevated risk of chronic liver disease (CLD) and hepatocellular carcinoma (HCC)[5,6]. Therefore, it is crucial to reveal the intricate regulatory mechanisms governing *A. flavus* development and secondary metabolite biosynthesis to control contamination caused by this pathogenic fungus.

Various factors affecting secondary metabolism and pathogenicity of *A. flavus* have been revealed in recent years, including environmental factors, transcriptional regulators, and histone post-translational modifications (HPTM)[7–9]. Nucleosome is the fundamental structural component of chromatin. Histones are involved in nucleosome assembly and thus influence chromatin packing[10]. Amino acid residues at the N-terminal-end of the histones are susceptible to post-translational modifications (PTMs), including acetylation, methylation, phosphorylation, and ubiquitination[11]. As a hotspot in epigenetic modification, research on HPTM has revealed its important role in *A. flavus* morphogenesis, secondary metabolite biosynthesis, and pathogenicity[12–14]. Methylation of H3K4, H3K36, and H3K79 is linked to transcriptional activation, whereas methylation of H3K9, H3K20, and H3K27 is linked to transcriptional repression and gene silencing[15]. In fungi, chromatin conformation is regulated by the methylation status of histone-associated sites, which in turn affects development and biosynthesis of secondary metabolites[16,17].

Methylation of lysine residues in histones is catalyzed by lysine (K) methyltransferases (KMTs)[18]. Set1, which contains a highly conserved SET (suppressor of variegation, enhancer of zeste, trithorax) domain, is the first and only H3K4 methyltransferase identified in yeast. The Set1-

[1]State Key Laboratory of Agricultural and Forestry Biosecurity, Key Laboratory of Pathogenic Fungi and Mycotoxins of Fujian Province, Key Laboratory of Biopesticide and Chemical Biology of Education Ministry, Proteomic Research Center, and School of Life Sciences, Fujian Agriculture and Forestry University, Fuzhou, China. [2]College of Animal Sciences, Fujian Agriculture and Forestry University, Fuzhou, China. ✉e-mail: wshyyl@sina.com

containing complex was named COMPASS (complex of proteins associated with Set1)[19–21]. The following studies have uncovered the catalytic activity of COMPASS toward H3K4 methylation[22–24]. In yeast, structure integrity of the COMPASS and methyltransferase activity of Set1 are maintained by Swd3, Swd1, Bre2, and Sdc1[25]. In yeast, Bre2 is the homolog of *Drosophila* Ash2 (absent, small, or homeotic 2-like protein) and functions as scaffold protein of COMPASS[22]. Available reports show that COMPASS is closely associated with meiotic DSBs (double-strand breaks)[26,27], and Bre2 is indispensable for H3K4me3 modification during yeast meiosis[28,29]. Bre2 and Sdc1 interact directly at the C-terminus of Bre2 and the Dpy-30 domain of Sdc1 in yeast. This interaction is required for appropriate H3K4 methylation and subsequent gene expression regulation[30]. In addition, CclA, the Bre2 ortholog in *A. nidulans*, was found to be related to the activation of cryptic secondary metabolites[31]. Thus, it is important to elucidate whether Bre2 plays a role in determining *A. flavus* virulence and AFB1 biosynthesis.

Arps (actin-related proteins) are essential functional subunits in the SWI/SNF, SWR1, and INO80 remodeling complexes. They play a critical role in regulating the ATPase activity of chromatin remodeler by binding to the helicase/SANT associated (HAS) domain of the remodeler at the N-terminal-end[32]. Among Arps, Arp9 binds to Sth1 and Snf2 of the SWI/SNF ATPases[32]. In *Saccharomyces cerevisiae*, the *arp9* mutant has impaired growth, a phenotype similar to that of the Swi-/Snf- mutant. Therefore, it was predicted that Arp9 functions in chromatin remodeling as a component of the remodeling complex rather than via enzymatic activity[33]. In *Penicillium oxalicum*, ARP9 plays an essential developmental role and is involved in the regulation of the expression of cellulase and amylase[34]. In *A. flavus*, Set1 mediates dimethylation and trimethylation of H3K4 and regulates the secondary metabolite and pathogenicity of *A. flavus*, which indicates that COMPASS plays a crucial role in regulating various important biological processes[35]. But the relationship between Bre2-mediated gene expression and Arp-mediated chromatin remodeling (especially Arp9) in secondary metabolite biosynthesis warrants further exploration.

In this work, Bre2 is chosen to explore its biological functions in fungal virulence and production of secondary metabolites, mainly AFB1. The role of Arp9 in the Bre2-mediated epigenetic signaling pathway is explored by ATAC-seq analysis. We elucidate the regulatory mechanism of Bre2-mediated H3K4 methylation and its potential role in fungal morphogenesis and secondary metabolite biosynthesis, thereby providing a theoretical basis for early prevention and control of contamination by *A. flavus* and its notorious secondary metabolites AFs.

## Results

### Bre2 plays a critical role in fungal morphogenesis and aflatoxin B1 synthesis

With the aim of exploring the role of Bre2 in fungal virulence from an evolutionary perspective, we identified *A. flavus* Bre2 (XP_002380746.1) via BLAST (the basic local alignment search tool) search using the amino acid (aa) sequence of *S. cerevisiae S288C* methyltransferase Bre2 (NP_013115.1) as bait against the NCBI database. According to bioinformatics analysis, *A. flavus* Bre2 consists of 617 aa and contains a conserved SPRY_Ash2 (SPRY) domain. Then, the orthologs of Bre2 from other 15 other model species were obtained through BLAST, and the phylogenetic tree was constructed using MEGA11.0 to analyze the evolutionary relationship between *A. flavus* Bre2 and these orthologs. The outcome demonstrated that filamentous fungi, especially *A. tanneri* (Identity: 79%), shared the closest evolutionary relationship with *A. flavus* Bre2 (Supplementary Fig. 1A). The domains of the Bre2 orthologs were analyzed with the NCBI database and visualized with DOG 2.0, which revealed that Bre2 is conserved among these model species; all Bre2 orthologs feature a highly conserved SPRY domain, and only *A. violaceofuscus* and human

possess additional domains: SAP130_C and PHD_ash2p_like (Supplementary Fig. 1B). To investigate the biological function of *A. flavus* Bre2, a deletion mutant (Δ*bre2*) and an in situ complementary strain (Com-*bre2*) were constructed, and the construction principles are illustrated in Supplementary Fig. 1C, D. The constructed Δ*bre2* and Com-*bre2* strains were confirmed by PCR, and the Δ*bre2* mutant was further confirmed by Southern blot (Supplementary Fig. 1E, F). The expression of *bre2* in both fungal strains was also monitored by RT-qPCR (Supplementary Fig. 1G). All the above results reflected that the Δ*bre2* and Com-*bre2* strains had been successfully constructed.

We then assayed biological function of Bre2 in *A. flavus* development and secondary metabolism. The WT, Δ*bre2*, and Com-*bre2* strains were inoculated on PDA and cultivated in the dark for 5 d at 37 °C. The results showed that the colony diameter of the Δ*bre2* strain was reduced to less than half that of WT strain, and it was fully restored in the Com-*bre2* strains (Fig. 1A, B). Further observation and statistical analysis of the conidia from the above fungal strains revealed that Δ*bre2* strain produced noticeably fewer conidia than the WT and Com-*bre2* strains (Fig. 1C). Subsequent RT-qPCR demonstrated that the expression levels of the key conidiation regulatory transcription factors *brlA* and *abaA* in the Δ*bre2* strain were significantly lower than those in the WT and Com-*bre2* strains, indicating that Bre2 regulates *A. flavus* conidiation via the classical sporulation pathway (Fig. 1D). To explore whether Bre2 is involved in sclerotium formation, the above *A. flavus* strains were inoculated onto CM (complete medium) at 37 °C for 6 d. The results showed that the Δ*bre2* strain could not produce any sclerotia, while the sclerotium formation ability was fully restored in the Com-*bre2* strain (Fig. 1E, F). In addition, RT-qPCR results showed that the expression levels of *nsdC* and *nsdD*, the key sclerotium formation transcription factors, in the Δ*bre2* strain were significantly lower than those in the WT and Com-*bre2* strains (Fig. 1G, H).

The effect of Bre2 on the biosynthesis of secondary metabolites was investigated by analyzing AFs production in the WT, Δ*bre2*, and Com-*bre2* strains via TLC (thin layer chromatography) and HPLC (high performance liquid chromatography). TLC results showed that AFs, including AFB1 synthesized by the Δ*bre2* strain, were significantly reduced compared to the WT strain, and the toxin production level was restored in the Com-*bre2* strain (Fig. 1I, J). The AFB1 production of each strain was further accurately quantified by HPLC, and the results also showed that the AFB1 yield of the Δ*bre2* strain was significantly lower than that of the WT and Com-*bre2* strains (Fig. 1K). We further explored the signaling pathway through which Bre2 modulates AFs production via assessing the transcript levels of aflatoxin biosynthesis-related genes. RT-qPCR analysis revealed that the expression levels of *aflR*, *aflS*, *aflC*, *aflD*, *aflK*, and *aflQ* in the Δ*bre2* strain were significantly downregulated compared with those in the WT and Com-*bre2* strains (Fig. 1L), indicating that Bre2 regulates AFB1 biosynthesis through the aflatoxin gene cluster. All the above results indicated that Bre2 significantly impacts fungal morphogenesis and the biosynthesis of the secondary metabolite AFB1 in *A. flavus*.

### The role of Bre2 in fungal colonization on hosts

The biological function of Bre2 in the colonization of *A. flavus* on crop hosts was further investigated. Peanut and maize kernels were inoculated with the spore suspensions (10⁵ spores/mL) of the WT, Δ*bre2*, and Com-*bre2* strains for 7 d. As shown in Fig. 2A–C, the conidial yield on both kernels inoculated with the Δ*bre2* spores was significantly lower than that on kernels inoculated with the WT and Com-*bre2* strains. The TLC showed that AFB1 production on both kernels inoculated with the Δ*bre2* strain was significantly lower than that on kernels inoculated with the WT strain (Fig. 2D–F). A further investigation into the role of Bre2 during fungal colonization on host crops was conducted by monitoring fungal virulence-related factors (including *cp1*, *fbg1*, *lysM*, *mhp1*, *mpg1* and *tox1*) via qRT-PCR, and the results showed that the absence of Bre2 significantly downregulated the expression of these

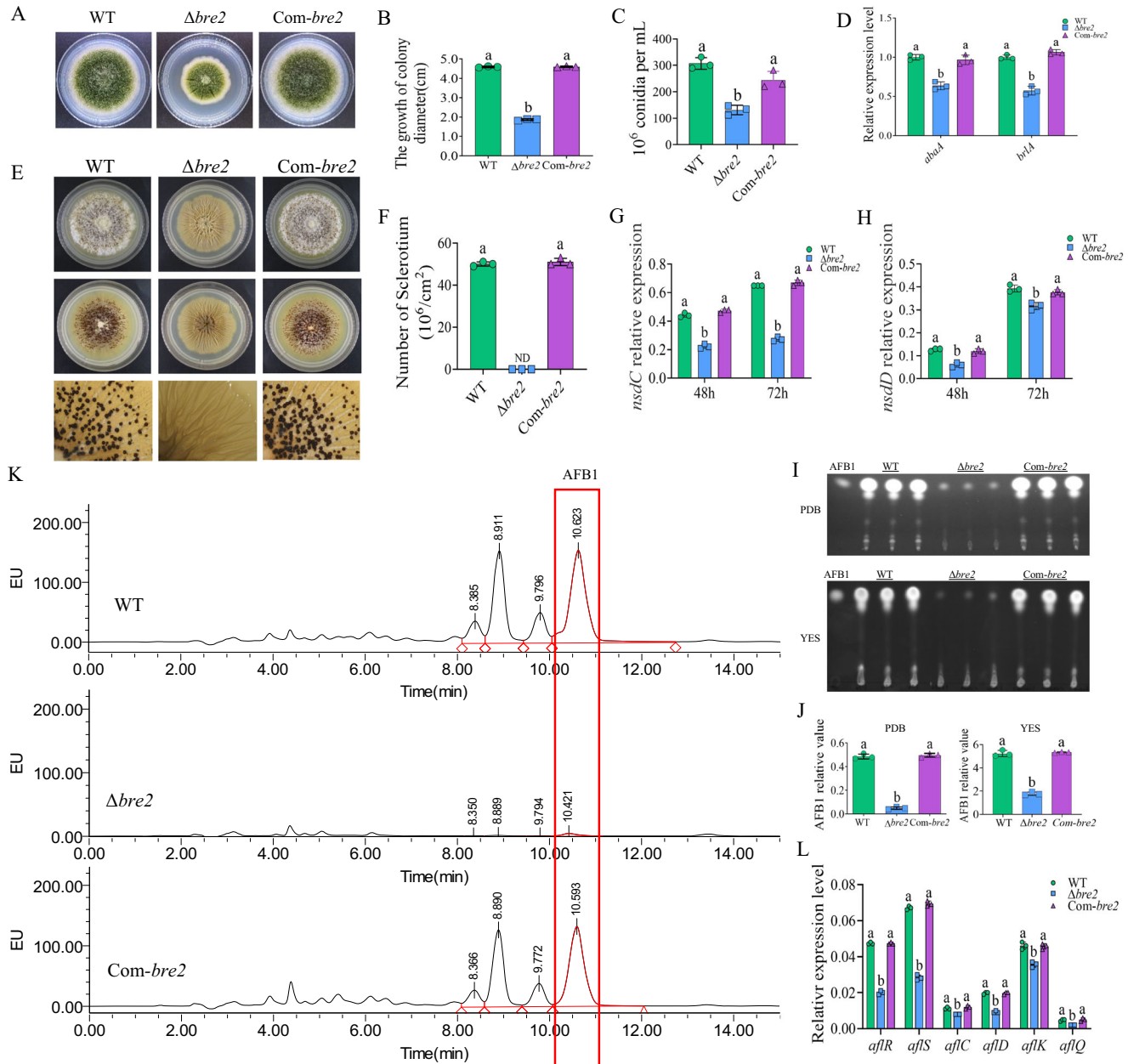

**Fig. 1 | Bre2 is involved in morphogenesis and aflatoxin production of A. flavus.**
**A** The colonies of WT, ΔBre2, and Com-Bre2 strains grown on PDA at 37 °C in dark for 5 d. **B** The statistics of colony diameter for the above fungal strains. **C** The statistics of spore production for the above fungal strains. **D** The transcriptional expression levels of *abaA* and *brlA* genes in WT, ΔBre2, and Com-Bre2 fungal strains at 48 h. **E** The WT, ΔBre2, and Com-Bre2 strains were point-inoculated on CM medium at 37 °C in dark for 7 d (above panels), then sprayed with 75% ethanol (middle panels), and the lower panels were the enlarged pictures of the middle panels under a dissecting microscope. **F** Statistics of sclerotium number for WT, ΔBre2, and Com-Bre2 strains. **G** Expression levels of the key sclerotium regulator genes *nsdC* in the formation of sclerotia in WT, ΔBre2, and Com-Bre2 strains at 48 h and 72 h. **H** The expression levels of the key regulator genes *nsdD* in the formation

of sclerotia in the above strains at 48 and 72 h. **I** TLC analysis of aflatoxin production in the above fungal strains. **J** The relative biosynthesis level of AFB1 from the above fungal strains was semi-quantified according to the results of (**I**). **K** The biosynthesis of aflatoxins in WT, ΔBre2, and Com-Bre2 strains was detected by HPLC after being grown on PDB at 29 °C in dark for 7 d. **L** Transcriptional levels of aflatoxin-synthesis related genes *aflR*, *aflS*, *aflC*, *aflD*, *aflK*, and *aflQ* from WT, ΔBre2, and Com-Bre2 strains. Data in (**B–D**, **F-H**, **J**, **L**) are presented as mean ± SD ($n = 3$). Bars in **B–D**, **F–H**, **J**, **L** with different top letters indicate significant differences ($P < 0.05$) based on One-way ANOVA and Tukey's multiple comparisons test, whereas those with the same letter mean no significant differences. ND means not detected. Source data are provided as a Source data file.

virulence factors (Supplementary Fig. 2A). The above results demonstrated that Bre2 plays an important promoting role in *A. flavus* colonization and AFB1 production on agricultural important crop hosts.

We then examined the effect of Bre2 in the infection of *Galleria mellonella* larvae. The larvae were injected with 8 μL ($10^7$ spores/mL) spores from the above fungal strains at the right hind proleg, respectively. The results demonstrated that the survival rate of larvae was

significantly higher in the Δ*bre2* mutant group compared with the WT and Com-*bre2* strain groups (Fig. 2G, H). To investigate the potential function of Bre2 in countering the host immune system, larval cellular immunity-related hemocytes and humoral immunity-related immune factors were monitored after injection with the aforementioned fungal strains. The results showed that following injection with Δ*bre2* spores, the hemocyte count in larvae decreased to approximately 60% of that

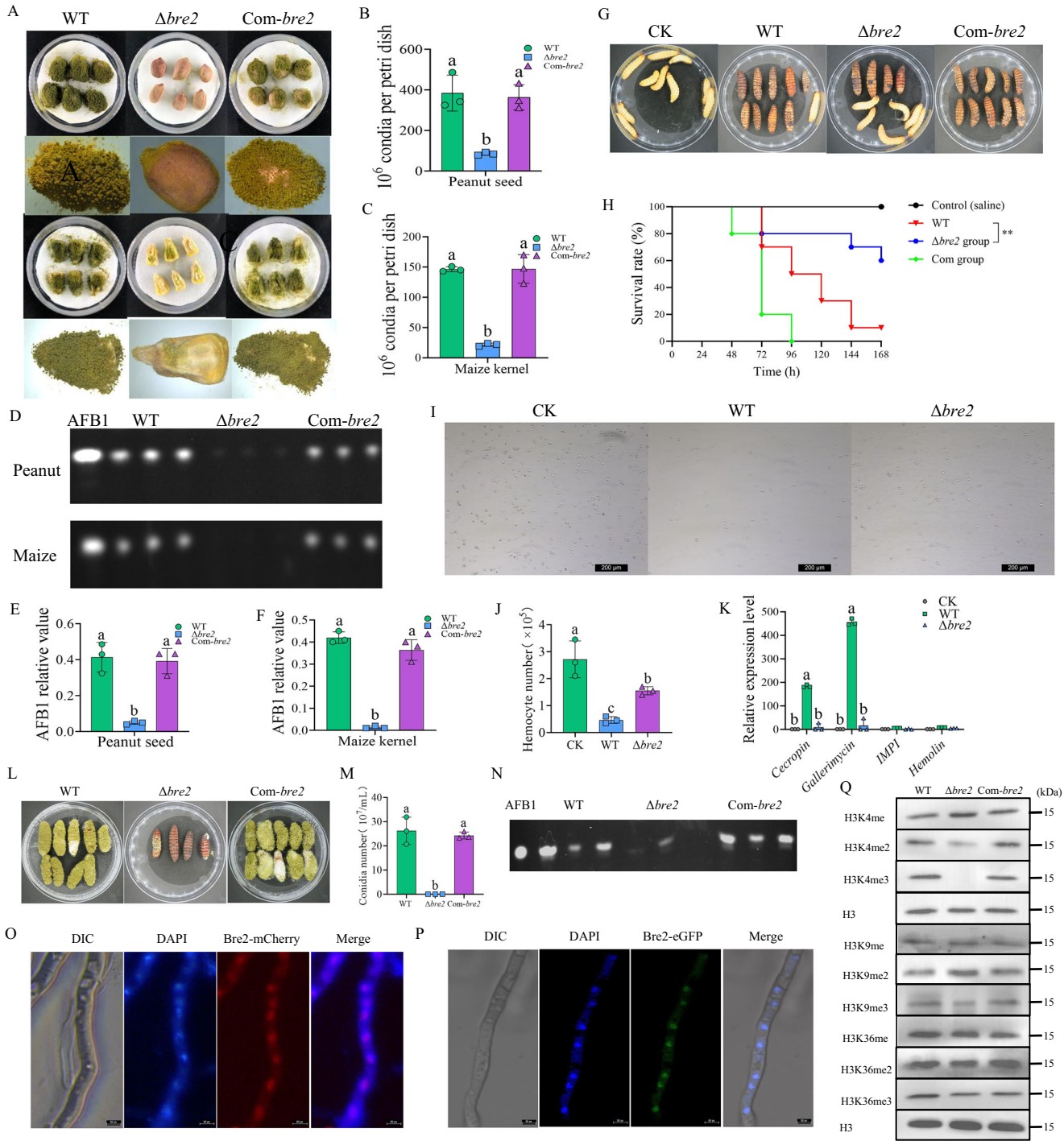

in the CK group. When injected with WT spores, it decreased to about 25% of that in the CK group, which was significantly lower than that in the Δ*bre2* injection group (Fig. 2I, J). Moreover, the expression levels of all immune factors, including *Cecropin*, *Gallerimycin*, *IMPI*, and *Hemolin*, were dramatically upregulated in the WT injection group, while the expression levels of these immune factors in the Δ*bre2* injection group showed no significant difference compared with the CK group (Fig. 2K). The sporulation capacity was significantly decreased in the infected group when the *bre2* gene was deleted, compared with the WT strain (Fig. 2L, M). Further TLC analysis showed that no AFB1 was detected in the Δ*bre2*-infected group (Fig. 2N). To evaluate the role of AFs in the virulence of *A. flavus* against animal hosts, each silkworm was injected daily with 5 μL 10% DMSO-dissolved AFB1 standard at three graded concentrations (0.05 mg/mL, 0.025 mg/mL, and

0.0125 mg/mL) for 7 d. The results showed that by the third day, the 0.05 mg/mL group exhibited an 80% mortality rate, the 0.025 mg/mL group showed a 60% mortality rate, and the 0.0125 mg/mL group exhibited only 10% mortality rate. Therefore, the 0.025 mg/mL concentration was selected for subsequent experiments (Supplementary Fig. 2B, C). Then, a new batch of silkworms was injected daily with 5 μL DMSO-dissolved AFs (with AFB1 maintained at approximately 0.025 mg/mL in the WT and Com-*bre2* strains) from the WT, Δ*bre2*, and Com-*bre2* strains. The results demonstrated that the survival rate of the Δ*bre2* group was significantly higher than that of Com-*bre2* and WT groups, particularly at 96 h and 120 h (Supplementary Fig. 2D, E), which reflected that the concentration of AFs is one of the critical factors in fungal virulence against hosts regulated by Bre2. A further exploration of the virulence determinant regulated by Bre2 was

**Fig. 2 | The role of Bre2 in regulation of _A. flavus_ virulence and histone methylation. A** Phenotypic analysis of corn and peanut kernels colonized by _A. flavus_ WT, Δ_Bre2_, and Com-_Bre2_ strains at 29 °C in dark for 7 d. **B** Statistical analysis of the conidia number of _A. flavus_ on the surface of peanut seeds. **C** Statistical analysis of the conidia number of _A. flavus_ on the surface of maize kernels. **D** TLC analysis of AFB1 yield in kernels infected by above fungal strains after 7 d inoculation. **E** Statistical analysis of AFB1 yield from peanut seeds colonized by the above fungal strains. **F** Statistical analysis of AFB1 yield from _A. flavus_ colonized maize kernels. **G** Photographs of the _Galleria mellonella_ larvae infected with the WT, Δ_Bre2_, and Com-_Bre2_ of _A. flavus_ strains after 1 week incubation. **H** The survival rate of _G. mellonella_ after 168 h (1 week) injection of the above strains, respectively (_n_ = 10). **I** The photos of hemocytes in the hemolymph of _G. mellonella_ larvae after injection with WT and Δ_Bre2_ strains. CK was injected with saline. **J** The hemocyte number of _G. mellonella_ larvae after infected with the above fungal strains, respectively. **K** The relative expression levels of the genes for immune-related

factors _Cecropin_, _Gallerimycin_, _IMPI_, and _Hemolin_. **L** Photographs of the dead larvae infected by _A. flavus_ after 7 d incubation in dark under 29 °C. **M** The histogram showing the conidia number on the larvae according to (**L**). **N** TLC analysis of AFB1 levels in larvae infected by above fungal strains after 7 d inoculation. **O** Subcellular location of Bre2 by Bre2-mCherry fusion expression strain. Results are representative of three independent experiments. **P** Subcellular localization of Bre2-eGFP fusion expression protein. Results are representative of three independent experiments. **Q** Western-blotting analysis on the role of Bre2 in the methylation of H3K4 (1-3), H3K9 (1-3), and H3K36 (1-3). Results are representative of three independent experiments. Data in (**B**, **C**, **E**, **F**, **J**, **K**, **M**) are presented as mean ± SD (_n_ = 3). Bars in **B**, **C**, **E**, **F**, **J**, **K**, **M** with different top letters indicate significant differences (_P_ < 0.05) based on One-way ANOVA and Tukey's multiple comparisons test, whereas those with the same letter mean no significant differences. Statistical analysis between groups in (**H**) was using the log-rank test. ** means significant difference of _P_ < 0.01. Source data are provided as a Source data file.

implemented via silkworm infection assays, in which silkworms were infected with 5 μL spore suspension ($1 \times 10^7$ spores/mL) from the Δ_bre2_, Com-_bre2_, and WT strains. The results showed that the survival rate of the Δ_bre2_ group was significantly higher than that of the WT and Com-_bre2_ groups (Supplementary Fig. 3A, B). The dead silkworms were collected and incubated at 29 °C in the dark for 7 d. Spore production was quantified using a hemocytometer, and fungal mycelial loading (reflecting hyphal growth) was assessed by qRT-PCR-mediated quantification of fungal DNA. The results demonstrated that, in contrast to the WT and Com-_bre2_ groups, the Δ_bre2_ group formed significantly fewer spores; however, it exhibited an increasing trend in mycelium production, although this trend was not statistically significant (Supplementary Fig. 3C–E). The above results indicated that Bre2 was involved in the regulation of fungal virulence of _A. flavus_ against animal hosts.

## Subcellular location and histone methylation targets of Bre2

The subcellular localization of Bre2 was determined by constructing Bre2-mCherry and Bre2-eGFP fusion expression strains. mCherry and eGFP were individually fused to the C-terminal of Bre2. The results showed that the red fluorescence of Bre2-mCherry and the green fluorescence of the Bre2-eGFP fusion protein were mainly detected in the nuclei (Fig. 2O, P), illuminating that Bre2 mainly aggregates in the nuclei to perform its biological function.

We then explored the function of Bre2 as histone methyltransferase by investigating its role in methylation of key histone lysine residues, including H3K4, H3K9, and H3K36, via Western blotting. The results showed that H3K4me3 was not detected in the Δ_bre2_ strain, and H3K4me2 was significantly downregulated, indicating that Bre2 is indispensable for H3K4me3 and an important factor in H3K4me2 modification (Fig. 2Q, up 4 lanes). We further explored whether Bre2 plays a role in the methylation of other key lysine residues in _A. flavus_ by assessing its catalytic activity toward H3K9 and H3K36. The results showed that Bre2 had no obvious regulatory function in the methylation of H3K9 and H3K36 (Fig. 2Q, lower 7 lanes). The above results revealed that Bre2 specifically regulates H3K4 methylation, exhibiting marked site specificity.

## The role of the SPRY domain and its key amino acids for the biological function of Bre2

As the most conserved domain in Bre2, the SPRY domain was detected in all selected species (Supplementary Fig. 1B). This domain is associated with a wide range of biological functions, including cytokine signaling and RNA metabolism[36–39]. We explored the role of the SPRY domain in the biological function of Bre2 by constructing the SPRY deletion mutant _bre2_$^{\Delta SPRY}$, followed by sequencing confirmation (Supplementary Fig. 4A). At the same time, the amino acid sequence of Bre2 orthologs were analyzed using MEME, and two most conserved amino acid residues, I343 and F349, were identified in the SPRY domain

(Supplementary Fig. 4B). To uncover the effect of these two conserved sites within the SPRY domain on the biological function of Bre2, we constructed the site mutants _bre2_$^{I343N}$ and _bre2_$^{F349S}$, which were confirmed by sequencing (Supplementary Fig. 4C). The role of the SPRY domain and its conserved amino acid residues I343 and F349 in H3K4 methylation was first analyzed with Western blotting, and the results showed that, like the Δ_bre2_ strain, no H3K4me3 was detected and H3K4me2 was significantly downregulated in the _bre2_$^{\Delta SPRY}$ strain. At the same time, both H3K4me3 and H3K4me2 were significantly downregulated in the _bre2_$^{I343N}$ and _bre2_$^{F349S}$ strains (Fig. 3A). The above results indicated that the SPRY domain is an indispensable core element of Bre2 for catalyzing the trimethylation and dimethylation of H3K4, and that I343 and F349 are the key amino acid residues responsible for Bre2's histone methylation-related catalytic function.

The biological effects of the SPRY domain and the conserved residues I343 and F349 were investigated by inoculating WT, _bre2_$^{\Delta SPRY}$, _bre2_$^{I343N}$ and _bre2_$^{F349S}$ strains on YES medium. Colony diameter was measured and the conidial number was counted after 4 d of incubation. The results (Fig. 3B–D) showed that the colony diameters of the _bre2_$^{\Delta SPRY}$, _bre2_$^{I343N}$, and _bre2_$^{F349S}$ were significantly smaller than those of WT, and their conidial numbers were only one-third of that of WT. The above results demonstrated that the SPRY domain is the core component of Bre2, and I343 and F349 are the key residues of this domain in regulating fungal colony growth and sporulation. To explore their role in sclerotium formation, the spores of the above fungal strains were inoculated on CM medium for 6 d at 37 °C. The results (Fig. 3E, F) showed that the _bre2_$^{\Delta SPRY}$, _bre2_$^{I343N}$, and _bre2_$^{F349S}$ could not form any sclerotia, indicating that the SPRY and its conserved residues I343 and F349 are indispensable for sclerotium formation. To evaluate the role of this domain in AFB1 biosynthesis, the above fungal strains were inoculated into PDB and cultured at 29 °C for 7 d, and the AFB1 production was detected by TLC and HPLC, respectively. The results showed that the AFB1 production ability of the _bre2_$^{\Delta SPRY}$, _bre2_$^{I343N}$, and _bre2_$^{F349S}$ strains in PDB was significantly lower than that of WT, and F349 exerted a more notable effect than I343 (Fig. 3G–I). This indicates that the SPRY domain and its conserved residues I343 and F349, especially F349, play a crucial role in regulating AFB1 biosynthesis.

Next, the colonization ability of the _bre2_$^{\Delta SPRY}$, _bre2_$^{I343N}$, and _bre2_$^{F349S}$ strains on maize kernels was further tested, and the results demonstrated that their colonization ability was significantly lower than that of the WT. The hyphae of these fungal mutants on maize kernel surfaces were markedly sparser than those of the WT, and their sporulation capacity was also dramatically reduced, especially for the _bre2_$^{\Delta SPRY}$ and _bre2_$^{F349S}$ strains (Fig. 3J, K). Further TLC detection revealed that the AFB1 levels detected from maize kernels inoculated with the above fungal mutant strains were significantly lower than those from the WT strain, especially in the _bre2_$^{\Delta SPRY}$ and the _bre2_$^{F349S}$ infection groups (Fig. 3L, M). All the above results confirm that the SPRY domain is a core component of Bre2, and its conserved residues I343 and F349 are

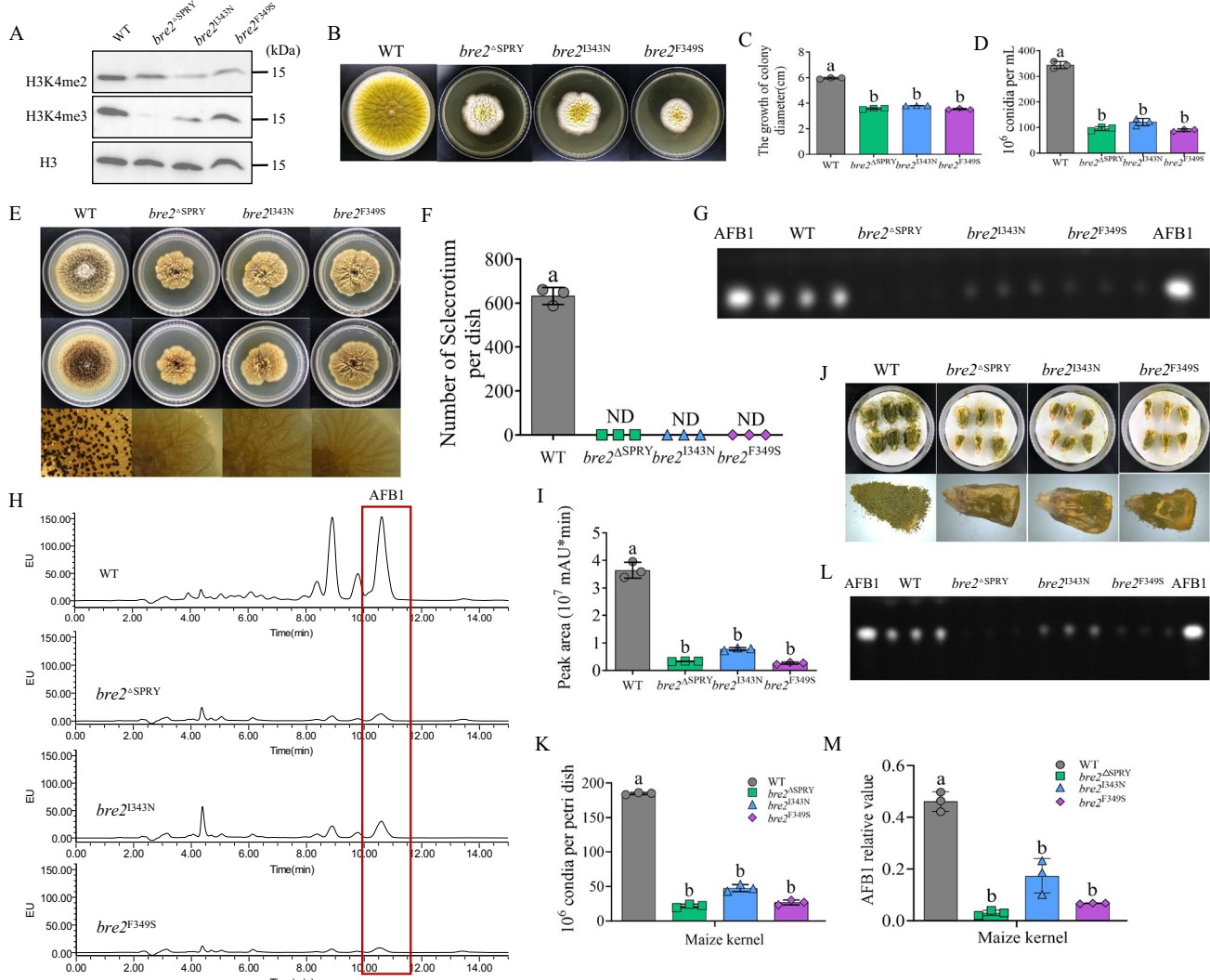

**Fig. 3 | Role of the SPRY domain and its I343 and F349 sites in H3K4 methylation and fungal colonization. A** Western blotting reflecting the role of SPRY domain and its I343 and F349 sites in the methylation modification of H3K4. Results are representative of three independent experiments. **B** The fungal strains, including WT, *bre2*^ΔSPRY, *bre2*^I343N and *bre2*^F349S were inoculated on YES medium under 37 °C in the dark for 4 d. **C** The statistics of colonial diameter for the above fungal strains. **D** The statistical histogram on the conidium number of the above fungal strains. **E** The WT, *bre2*^ΔSPRY, *bre2*^I343N and *bre2*^F349S strains were inoculated on CM media under 37 °C in the dark for 6 d. **F** The statistical histogram on the sclerotium number of the above fungal strains. **G** The AFB1 yield of WT, *bre2*^ΔSPRY, *bre2*^I343N, and *bre2*^F349S strain were monitored by TLC after cultured at 29 °C for 7 d in the dark. **H** HPLC analysis of aflatoxin yield from above fungal strains. **I** Statistical

results of HPLC peak area of AFB1 from the above fungal strains. **J** The colonization of WT, *bre2*^ΔSPRY, *bre2*^I343N, and *bre2*^F349S strains on the surface of maize kernels at 29 °C for 7 d in the dark. **K** The statistical analysis of the conidia number produced by the above fungal strains on the maize surface. **L** TLC analysis of AFB1 yield on these maize kernels colonized by above fungal strains. **M** Statistical analysis of AFB1 content produced by above strains colonized on maize kernels. Data in (**C**, **D**, **F**, **I**, **K**, **M**) are presented as mean ± SD ($n = 3$). Bars in **C**, **D**, **F**, **I**, **K**, **M** with different top letters indicate significant differences ($P < 0.05$) based on One-way ANOVA coupled with Tukey's multiple comparisons test, whereas those with the same letter mean no significant differences. ND means not detected. Source data are provided as a Source data file.

key elements of the SPRY domain for mediating the biological functions associated with H3K4me3 and me2 modification.

## Characterization of H3K4me3 modified chromatin fragment by ChIP-seq

We investigated the intrinsic molecular mechanism by which Bre2 regulates aflatoxin synthesis and fungal virulence via H3K4me3 using ChIP-seq by anti-H3K4me3 monoclonal antibody. Genes associated with these fragments in the WT were enriched and sequenced, with the Δ*bre2* strain serving as a control to exclude non-specific and false-positive signals. Clean reads from sequencing were aligned to the genome of *A. flavus* using BWA software, and the read density distribution map was generated by Deeptools software (version: 2.5.4). The results (Supplementary Fig. 5A) showed that reads of H3K4me3-

modified chromatin fragments from the WT IP samples were highly enriched in the TSS region. Principal component analysis (PCA) (Supplementary Fig. 5B) and the sample correlation analysis heat map (Supplementary Fig. 5C) showed that the above samples met the quality requirements for subsequent research. The differentially accumulated peaks (DAPs) were distributed across the entire *A. flavus* genome (Supplementary Fig. 5D), and the results in Supplementary Fig. 5E showed that the most enriched peaks in the WT were localized to exons and promoter regions, which were further confirmed by the results of the combined analysis (Supplementary Fig. 5F). The above findings indicated that the H3K4me3 modification mediated by Bre2 in *A. flavus* exhibits extensive genome-wide distribution and is primarily targeted to chromatin regions encompassing promoters and exons, which are closely associated with gene expression and regulation.

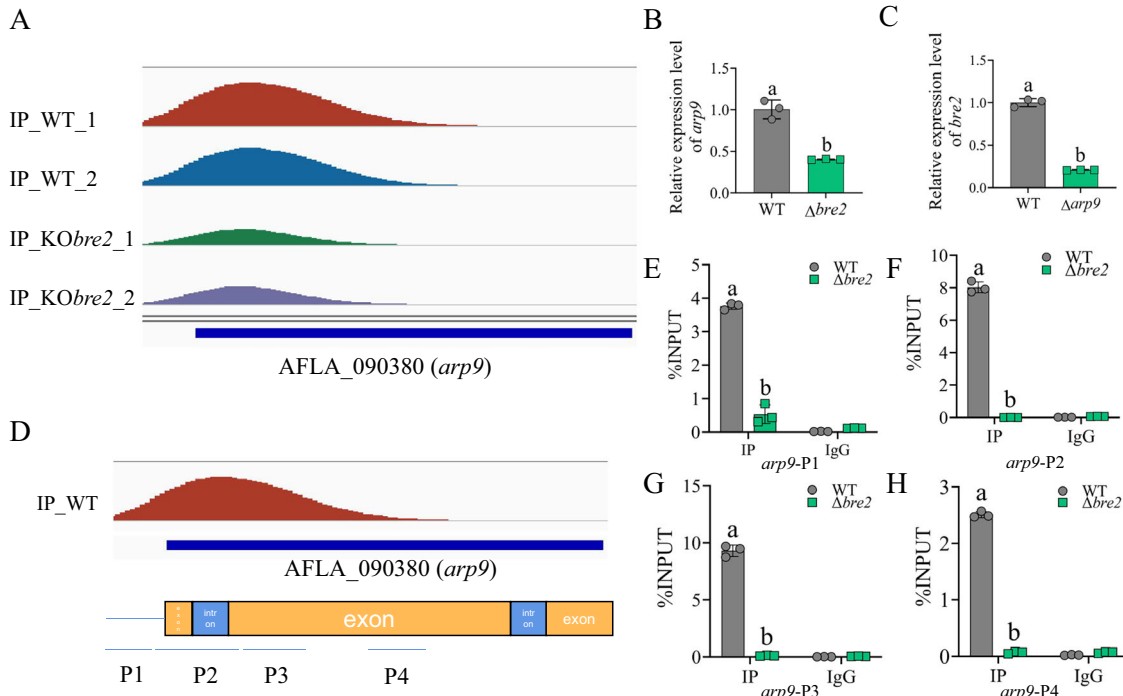

**Fig. 4 | Bre2 mediates H3K4me3 modification of arp9. A** Comparison of the enriching levels of the H3K4me3-modified chromatin fragment of *Arp9* gene between WT and Δ*bre2* strains. **B** RT-qPCR analysis on the expression level of the *arp9* gene in WT and Δ*bre2* strains. **C** RT-qPCR analysis on the expression level of the *bre2* gene in WT and Δ*arp9* strains. **D** The ChIP-qPCR analysis for H3K4me3 modification on *arp9*. P1-P4 represents the monitored fragments on the *arp9*. **E**–**H** The ChIP-qPCR results of P1 to P4, verifying the modification of H3K4me3

mediated by Bre2. IP means chromatin immunoprecipitation samples, while IgG is random antibody as the negative immunoprecipitation control. Data in (**B**, **C**, **E**–**H**) are presented as mean ± SD ($n = 3$). Bars in **B**, **C**, **E**–**H** with different top letters indicate significant differences ($P < 0.05$) based on the unpaired two-tailed *t*-test, whereas those with the same letter mean no significant differences. Source data are provided as a Source data file.

The ChIP-seq analysis identified 2871 up-modified and 723 down-modified DAPs in the WT compared to the Δ*bre2* strain, as shown in Supplementary Fig. 5G. To further explore the role of H3K4me3 modification in genetic regulation at the genomic level, the up-modified DAP-associated genes in the WT were subjected to GO annotation. Compared with the Δ*bre2* strain, these highly enriched up-modified genes in the WT were widely involved in various biological processes, including the mitotic cell cycle, regulation of cellular processes, and regulation of biological processes, etc. In the cellular component, these DAP-associated genes in the WT were mainly enriched in nuclei and chromosomes, and in the molecular function category, they were only found to be associated with the biological function of binding (Supplementary Fig. 5H). KEGG analysis of up-modified DAP-associated genes of the WT related to the Δ*bre2* strain showed that these genes were heavily involved in multiple biological processes, including lipid metabolism, carbohydrate metabolism, amino acid metabolism, translation, transcription, folding, sorting and degradation, transport and catabolism, cell growth and death, information processing, as well as metabolism of terpenoids and polyketidase (Supplementary Fig. 5I). The above results indicated that H3K4me3 modification catalyzed by the COMPASS complex, with Bre2 as a key subunit, plays an indispensable role in fungal growth and development, stress responses, and secondary metabolites production in *A. flavus*.

### Arp9 is the key target regulated by Bre2-mediated H3K4me3

The chromatin remodeling protein (CRP) family includes a diverse set of enzymes with distinct biological functions. In recent years, the SNF2 family, as a representative member of the CRP family, has been found to be profoundly involved in various kinds of biological processes[40–42]. In this study, we comprehensively mined information from DAP-

associated genomic data and found that the H3K4me3 modification-related genes include several SNF2 family genes harboring DEAQ-box helicase domain. To reveal the role of the SNF2 family in Bre2-mediated biological regulatory mechanisms underlying fungal morphology and virulence, we further investigated Arp9–an important subunit of the SNF2 family–to unveil how Bre2 modulates diverse fungal biological processes through Arp9 via H3K4me3 modification.

The enrichment level of H3K4me3 modification at the *arp9* coding region of the WT and Δ*bre2* strains was analyzed and visualized using IGV 2.3.91 based on ChIP-seq data. These results showed that the enrichment peaks were located at the *arp9* promoter and 5′ end exon regions, and the enrichment level (quantified by peak areas) of this chromatin fragment in the WT was $2^{1.1}$-fold higher than that in the Δ*bre2* strain (FDR < 0.05) (Fig. 4A). RT-qPCR analysis showed that the expression level of *arp9* in the WT was nearly threefold higher than that in the Δ*bre2*, indicating that Bre2 positively regulates *arp9* expression via H3K4me3 modification (Fig. 4B). Further RT-qPCR analysis showed that the expression level of *bre2* in the Δ*arp9* strain was approximately fivefold lower than that in the WT strain, indicating that Arp9 dramatically promotes *bre2* expression and forms a positive feedback loop with Bre2-mediated epigenetic regulation (Fig. 4C).

We examined whether Bre2 regulates Arp9 through catalyzing H3K4me3 by performing ChIP-qPCR analysis on WT and Δ*bre2* strains. The detected fragments were located from P1 to P4 as shown in Fig. 4D, covering the promoter (P1), 5′ exon fragments (P2 and P3), and middle exon fragment (P4). ChIP-qPCR analysis revealed that the enrichment of P1 reached approximately 4% of the INPUT, while that at both P2 and P3 reached about 8%; enrichment at P4 in the middle exons was only about 2.5%. These results reflected that Bre2 mediates H3K4me3 modification of *arp9* primarily at its promoter and 5′ exons, confirming

that Bre2 upregulated *arp9* expression via H3K4me3 modification at these regulatory regions (Fig. 4E–H).

## Arp9 plays an important role for A. flavus development and virulence

Arp9 is an essential component of chromatin remodeling and modifying complex (SWI/SNF complex). We constructed *Arp9* deletion mutant (Δ*Arp9*) and complementary strain (Com-*Arp9*) and validated by PCR and RT-qPCR (Supplementary Fig. 6A, B). To determine the subcellular localization of Arp9, an Arp9-eGFP fusion fungal strain was constructed. The findings demonstrated that the green fluorescence-tagged Arp9-eGFP accumulated precisely at the nuclear location (Supplementary Fig. 6C), reflecting that Arp9 is predominantly localized to the nucleus to exert its biological functions.

Phenotypic analysis revealed that Arp9 significantly improves colonial growth by enhancing spore germination, dramatically increases sporulation via the AbaA- and WetA-mediated conidiation pathway (Supplementary Fig. 6D–I), induces sclerotia formation through the NsdC- and NsdD-mediated sclerotium formation pathway (Supplementary Fig. 6J–L), and is nearly indispensable for AFB1 biosynthesis by upregulating the transcript levels of aflatoxin-related genes, including *aflR*, *aflS*, and *aflP* (Supplementary Figs. 6M and S6N). The biological impact of Arp9 on fungal colonization was investigated by inoculating crop kernels with the spore suspensions ($10^7$ spores/mL) prepared from the WT, Δ*arp9*, and Com-*arp9* strains, respectively. The results showed that the conidial yield on crop hosts infected with Δ*arp9* (Supplementary Fig. 7A) was significantly lower than that of the WT group (Supplementary Fig. 7B, $P < 0.001$). TLC analysis showed that almost no AFB1 was detected in the crop hosts infected with Δ*arp9* (Supplementary Fig. 7C). The above findings demonstrate that Arp9 is critical for the pathogenicity of *A. flavus*.

## Arp9 interacts with RSC8, Arp7 and Sth1

The interaction between Arp9 and other subunits of the SWI/SNF complex was investigated using immunoprecipitation coupled with mass spectrometry (IP-MS). First, Arp9 was HA-tagged, and the successfully tagged strain was confirmed by Western blotting. Immunoprecipitation was performed to enrich Arp9-interacting proteins, and the enrichment efficiency was confirmed by Western blotting, as shown in Fig. 5A. Subsequently, the samples were analyzed by mass spectrometry. The base peak chromatograms were obtained (Fig. 5B), indicating that the tested samples exhibited high separation resolution and signal intensity and were qualified for further identification. The identified candidate interacting proteins are listed in Supplementary Data 1. The results showed that a total of 63 proteins were pulled down by Arp9. Through joint analysis with the STRING database (a tool for predicting functional protein association networks; string-db.org), the Arp9-interacting protein RSC8 was screened out from these 63 candidates based on the criterion that its enrichment level in the Arp9-HA strain was fourfold higher than that in the WT strain. Like Arp9, RSC8 is a subunit of the SWI/SNF complex[43]. It has been reported that, within the SWI/SNF complex of *S. cerevisiae*, Arp9 first forms a heterodimer with another subunit, Arp7, before further associating with the core subunit Sth1 to mediate chromatin remodeling[32]. To explore the direct interaction between RSC8 and Arp9, and to verify the interactions of Arp9 with Arp7 and Sth1 in the *A. flavus* SWI/SNF complex, Co-IP analysis was performed. The dual-tagged *A. flavus* strains (the Arp9-3HA strain, in which RSC8, Sth1, and Arp7 were further labeled with an eight-amino-acid Strep tag that specifically binds to streptavidin or its derivative StrepTactin, respectively) were verified by sequencing. Subsequent Western blotting verified the successful fusion of the Strep tag to RSC8, Sth1, and Arp7 in the Arp9-3HA background: two transformants of the Arp9-3HA/RSC8-Strep strain (Fig. 5C), four transformants of the Arp9-3HA/Sth1-Strep strain (Fig. 5D), and five transformants of the Arp9-3HA/Arp7-Strep strain (Fig. 5E). Subsequent

Co-IP experiments using the Arp9-3HA/RSC8-Strep strain showed that Arp9-3HA was successfully pulled down, enriched and detected with an anti-HA antibody, further detection with an anti-Strep antibody revealed that RSC8 interacted with Arp9 and was co-precipitated with Arp9 when anti-HA antibody was used for pull-down (Fig. 5F upper panel). The lower panel of Fig. 5F revealed that in the same fungal strain, RSC8-Strep could be specifically pulled down, enriched, and detected with an anti-RSC8 antibody. Meanwhile, anti-HA antibody-based detection validated the interaction between Arp9 and RSC8, as Arp9 was co-pulled down with RSC8 by the anti-Strep antibody. Consistent with the aforementioned approach, Arp9 was also found to interact directly with the SWI/SNF complex subunit Arp7, as depicted in Fig. 5G. Notably, immunoprecipitation of Arp9 with anti-HA magnetic beads from the Arp9-3HA/Sth1-Strep strain yielded no detectable Sth1-Strep in the subsequent Co-IP analysis with an anti-Strep antibody (Fig. 5H, upper two panels). Conversely, initial immunoprecipitation of Sth1 using anti-Strep magnetic beads resulted in the detection of Arp9 via anti-HA antibody (Fig. 5H, lower two panels). The findings presented in Fig. 5H lead to the hypothesis that in *A. flavus*, Arp9 forms a heterodimer with Arp7 to mediate its interaction with the core subunit Sth1, and that the interaction of Arp9 alone with Sth1 is substantially weaker than that between Sth1 and the Arp9-Arp7 heterodimer.

## Arp9 collaborates with Bre2 to modulate fungal morphogenesis and secondary metabolism

To explore regulatory mechanism of Bre2 and Arp9 in fungal development and AFB1 biosynthesis, we constructed double knockout strain Δ*arp9*Δ*bre2* via homologous recombination (Fig. 6A). The fungal strains, including the Δ*bre2*, Δ*arp9*, Δ*arp9*Δ*bre2*, and WT strains, were incubated on PDA for 4 d. Subsequently, the colonial diameters and spore number were measured and calculated. The results showed that the colony diameter of Δ*arp9*Δ*bre2* was significantly smaller than those of single knockout strains Δ*bre2* and Δ*arp9*, and far smaller than that of the WT strain (Fig. 6B, C). The spore number data showed a similar pattern to that of colonial diameter. The conidia number of Δ*arp9* and Δ*arp9*Δ*bre2* were both dramatically lower than that of the Δ*bre2* strain, suggesting that Arp9 might act downstream of Bre2 in the signaling pathway. No sclerotia were found in Δ*bre2*, Δ*arp9*, and Δ*arp9*Δ*bre2* strains (Fig. 6E, F), indicating that Bre2 and Arp9 are both indispensable for sclerotium formation in this signaling pathway. TLC analysis showed that no obvious AFB1 was detected in the Δ*bre2*, Δ*arp9*, and Δ*arp9*Δ*bre2* strains. Collectively, these results suggested that Bre2 and Arp9 are deeply involved in hyphal growth and conidiation, are indispensable for sclerotia formation and AFB1 biosynthesis, and that Arp9 might act downstream of Bre2 in the Bre2-triggered epigenetic signaling pathway. Additionally, they may establish a positive feedback loop, given that Arp9 dramatically promotes the expression of *bre2* (Fig. 4C).

## Arp9 plays a key role in chromatin remodeling

Arp9 plays a key role in adjusting the expression levels of downstream regulatory genes by remodeling chromatin accessibility, which is crucial for elucidating the molecular mechanisms by which Arp9 regulates the virulence of *A. flavus*, particularly the production of secondary metabolites such as aflatoxins. Therefore, Δ*Arp9* and WT strains were used to perform ATAC-seq (Assay for transposase-accessible chromatin using sequencing). The results revealed that the correlation coefficient between each sample was greater than 0.8 (Fig. 7A), indicating that the sample stability met the requirements and was qualified for subsequent operations. Through differential expression analysis, we found that compared with WT, the number of downregulated differentially expressed genes (DEGs) in Δ*Arp9* strain was dramatically higher than that of upregulated DEGs (214 vs 22), indicating that Arp9 affects chromatin accessibility mainly by promoting chromatin opening (Fig. 7B). The 214 downregulated DEGs of the Δ*Arp9* strain were

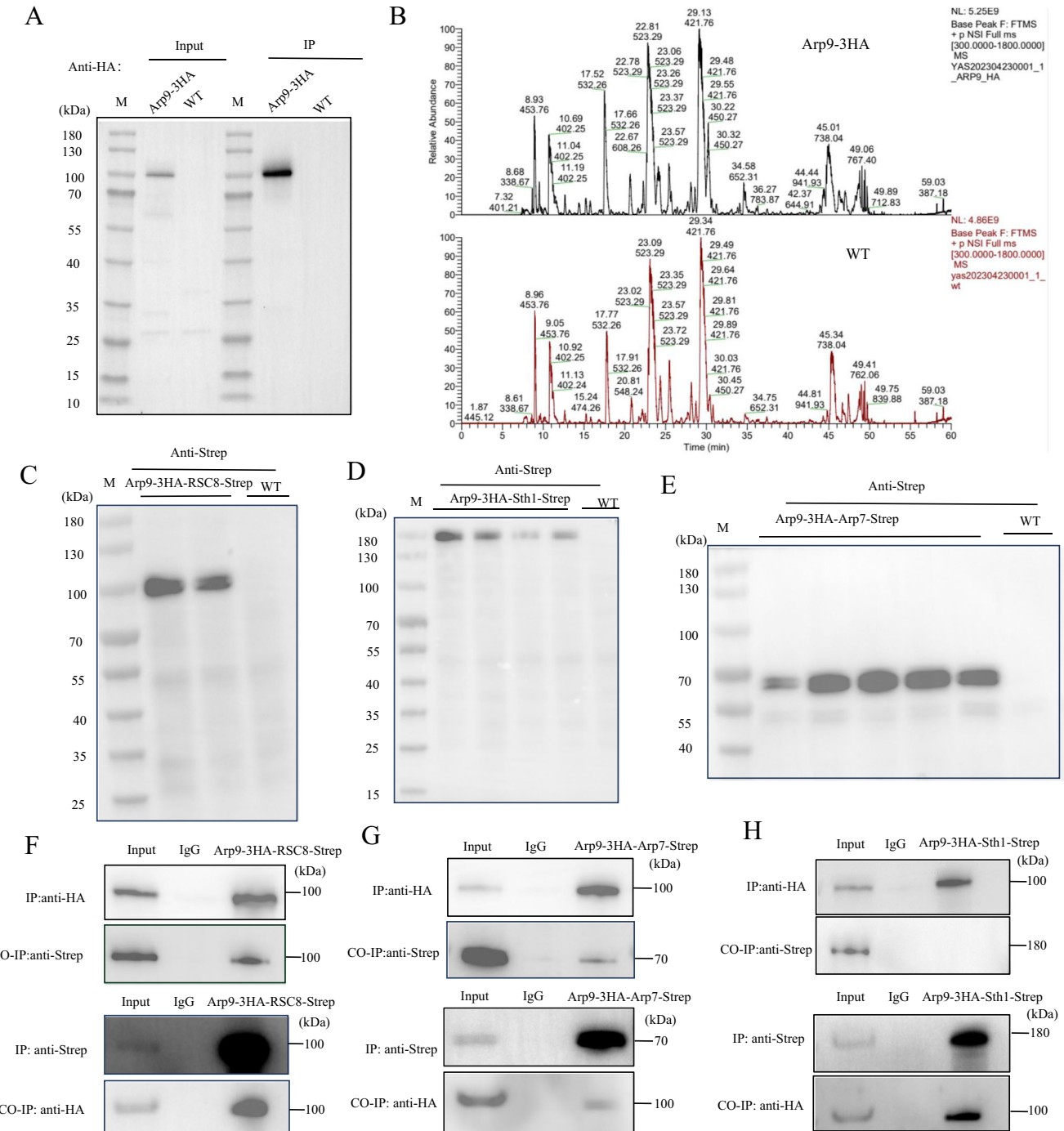

**Fig. 5 | Co-IP analysis-based interaction assay betweenArp9 and RSC8, Arp7 or Sth1. A** The protein bands of Arp9-3HA strain and WT strain were detected by western-blotting assay. Lane M: molecular weight marker. Results are representative of three independent experiments. **B** The results of mass spectrometry from immunoprecipitation of the Arp9-3HA and WT strain. The target proteins obtained from immunoprecipitation were digested with trypsin, and the peptide segments were analyzed using LC-MS/MS (nanoLC QE) after enzymatic hydrolysis. The data was collected in positive ion mode, and the ion chromatogram (the peak chromatogram) was obtained by full MS scanning (*m/z* range 300–1800). **C–E** Western blotting was carried out to verify the constructed Arp9-3HA/RSC8-Strep, Arp9-3HA/Arp7-Strep, and Arp9-3HA/Sth1-Strep strains. Lane M: molecular weight marker. Results are representative of three independent experiments. **F–H** Co-IP explored the interaction between Arp9 and RSC8, Arp7, and Sth1. Results are representative of three independent experiments. **F** The upper panel Western-blotting results were obtained from the Arp9-3HA/RSC8-Strep strain: after IP with

anti-HA magnetic beads, Arp9 (100 kDa) was detected using an anti-HA antibody, and RSC8 (100 kDa) using an anti-Strep antibody. The lower panel shows IP with anti-Strep beads, followed by western blotting detection of RSC8 using an anti-Strep antibody, and Arp9 using an anti-HA antibody. **G** The upper panel shows western blotting results from the Arp9-3HA/Arp7-Strep strain: after IP with anti-HA magnetic beads, Arp9 was detected using an anti-HA antibody, and Arp7 (70 kDa) using an anti-Strep antibody. The lower panel was IP with anti-Strep beads, followed by western blotting detection of Arp7 via an anti-Strep antibody, and Arp9 via an anti-HA antibody. **H** The upper panel shows western blotting results from the Arp9-3HA/Sth1-Strep strain: after IP with anti-HA magnetic beads, Arp9 was detected using an anti-HA antibody, and Sth1 (180 kDa) using an anti-Strep antibody. The lower panel presents IP with anti-Strep beads, followed by western-blotting detection of Sth1 via an anti-Strep antibody, and Arp9 via an anti-HA antibody. Source data are provided as a Source data file.

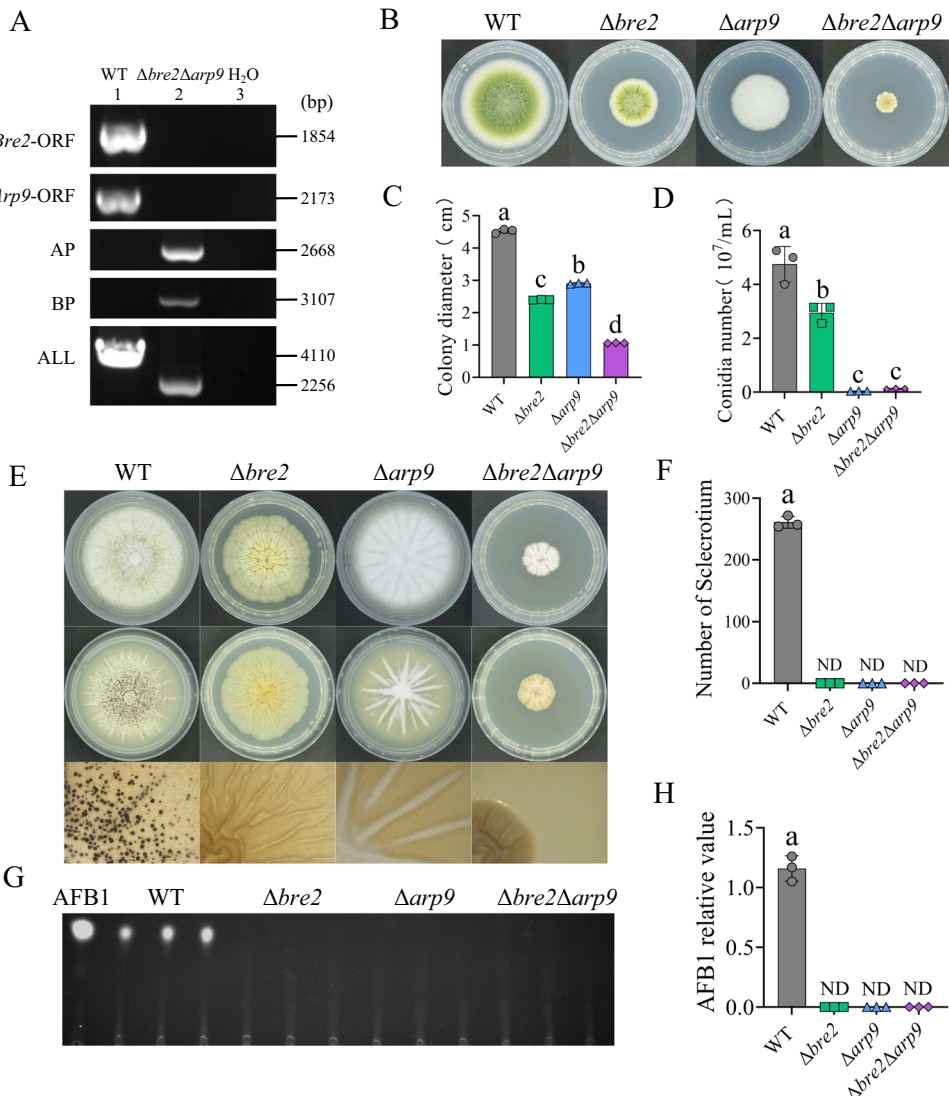

**Fig. 6 | Arp9 collaborates with Bre2 to regulate fungal development and AFB1 biosynthesis. A** The double knockout strain Δarp9Δbre2 was verified through diagnostic PCR. Compared to WT strain, the bre2 ORF and arp9 ORF could not be amplified from the Δbre2Δarp9 strain; while AP and BP fragments could be amplified from Δbre2Δarp9 strain, but could not be detected from WT. The DNA fragments amplified from WT is bigger than that form the Δbre2Δarp9 strain with nested primers Bre2-pyrG-NF and Bre2-pyrG-NR. The AP fragment, which overlaps the 5' flanking sequence and pyrG, was amplified with the validation primers Arp9-AP-F and Arp9-AP-R. The BP fragment, overlapping pyrG and the 3' flanking sequence, was amplified by primers Arp9-BP-F and Arp9-BP-R. Results are representative of three independent experiments. **B** The spore of Δbre2, Δarp9, Δarp9Δbre2, and WT strains (1 μL, 10⁷spores/mL) were incubated on PDA for 4 d.

**C** The diameter of the above fungal strains was calculated according to the (**B**). **D** The conidia number of the above fungal strains was calculated according to the (**B**). **E** The spore of Δbre2, Δarp9, Δarp9Δbre2 and WT strains (1 μL, 10⁷spores/mL) were incubated on CM for 7 d. **F** The statistic result of sclerotia number of the above fungal strains according to the (**E**). **G** TLC analysis of the AFB1 yield in the above fungal strain. These fungi were grown in PDB, at 29 °C for 7 d. **H** The relative quantitative of AFB1 yield in the above fungal strains according to the (**G**). Data in (**C, D, F, H**) are presented as mean ± SD ($n = 3$). Bars in **C, D, F, H** with different top letters indicate significant differences ($P < 0.05$) based on One-way ANOVA coupled with Tukey's multiple comparisons test, whereas those with the same letter mean no significant differences. ND means not detected. Source data are provided as a Source data file.

evenly distributed across each chromosome (Fig. 7C), of which 63.84% were located in the promoter region, indicating that Arp9 mainly promotes chromatin opening in the promoter region of target genes (Fig. 7D). Subsequently, GO and KEGG analysis were performed to identify the main physiological functions and signaling pathways of genes associated with these 214 down-regulated DEGs. Among the GO annotations, these DEGs were involved in the regulation of metabolic processes, response to stimulus, biological regulation, localization, and other physiological functions (Fig. 7E). Further KEGG pathways analysis showed that these downregulated DEGs in the ΔArp9 mutant were heavily enriched in metabolism process (accounting for 60% genes of KEGG-annotated genes (the number of genes in the metabolism related KEGG pathway divided by the total number of KEGG

annotated genes), and 73% GO-annotated genes), including lipid metabolism, carbohydrate metabolism, amino acid metabolism, nucleotide metabolism, metabolism of terpenoids and polyketides, and biosynthesis of other secondary metabolites (Fig. 7F). The above results indicated that Arp9 regulates chromatin remodeling by promoting chromatin accessibility at promoter regions and participates in various important regulatory processes, particularly those related to oxidation and secondary metabolism.

## Arp9 regulates fungal virulence through downstream targets by chromatin accessibility regulation

Further dissection of the signaling pathway mediated by Arp9 was performed by filtering 214 downregulated genes according to the

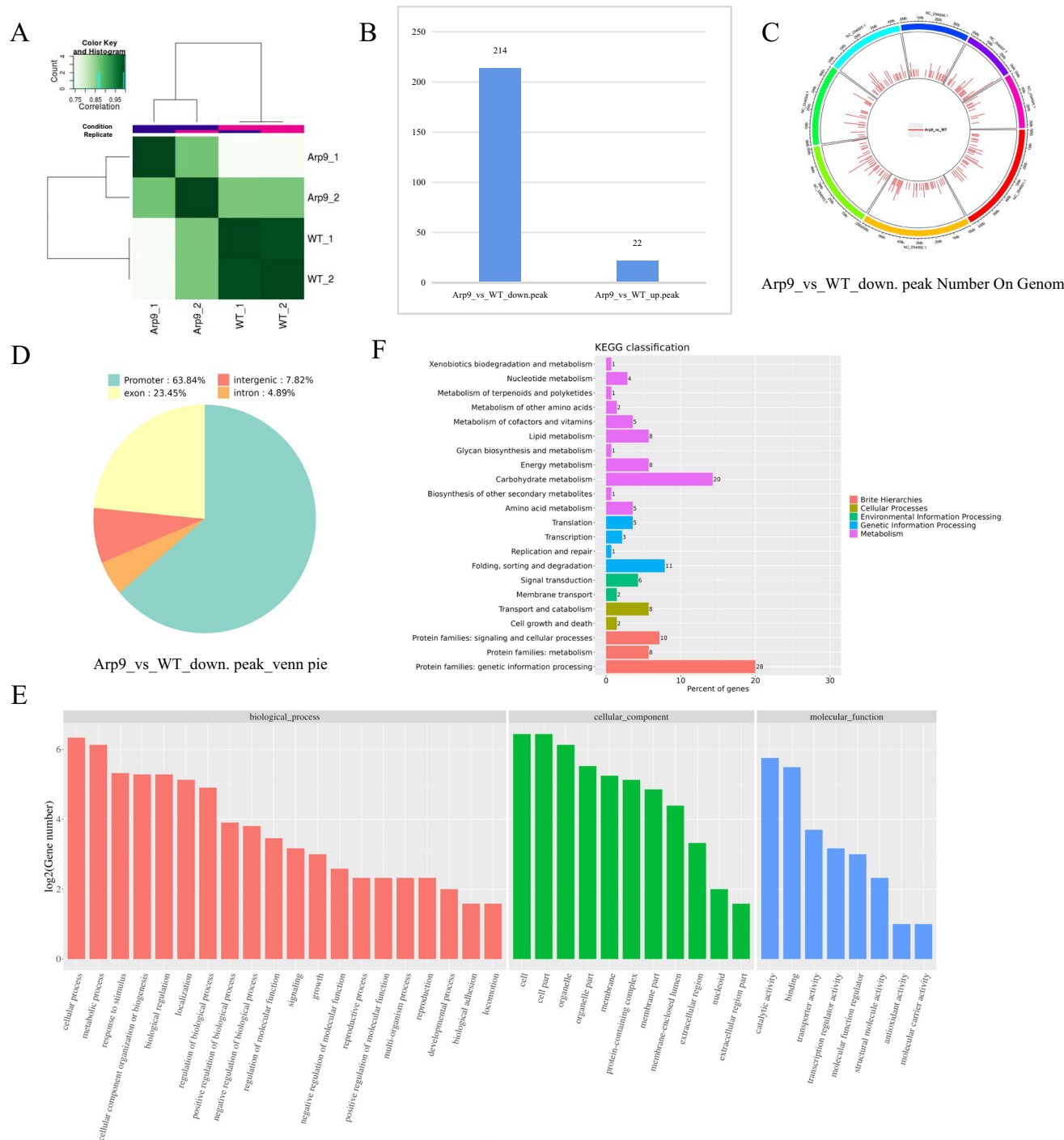

**Fig. 7 | Analysis of ATAC-seq sequencing data. A** Correlation analysis of ATAC-seq fungal samples. **B** The histogram statistics of DAPs distribution by comparing Δ*Arp9* to WT strains. **C** The distribution of 214 down-regulated DAPs of Δ*Arp9* on the *A. flavus* genome. **D** The Venn pie showing the distribution of the down-regulated DAPs in the functional elements of *A. flavus* genome. **E** GO cluster analysis of the down-regulated DAPs related genes. **F** The KEGG pathways analysis on the down-regulated DAPs. Arp9 in this figure stands for the Δ*Arp9* strain.

molecular function-based GO enrichment. Through this filtering, genes related to fundamental molecular-level activities (such as catalysis or binding) as well as those involved in the ordered biochemical or physiological processes at the cellular, tissue, or organismal level (listed under biological process) were screened out (*P* < 0.05). Finally, nine candidate genes were identified in the second round of selection via KEGG pathway annotation of genes implicated in growth/development, reproduction, virulence, autophagy, proliferation, cell wall integrity, drug resistance, and immunity. These candidate genes

include stromal membrane-associated protein (SMAP, G4B84_002937), phosphatidylinositol 4-kinase A (PIKA, G4B84_005089), phosphatidylinositol 4-kinase type 2 (PIK2, G4B84_005528), mannitol 2-dehydrogenase (M2DH, G4B84_010461), and phosphatidylserine decarboxylase (PSD, G4B84_011294), MFS transporter (UMF1, G4B84_002788), ATP-binding cassette (CDR1, G4B84_009741), PH and SEC7 domain-containing protein (PSDP, G4B84_009537), and triose/dihydroxyacetone kinase/FAD-AMP lyase (DAK (TKFC), G4B84_008657). The above genes were first monitored

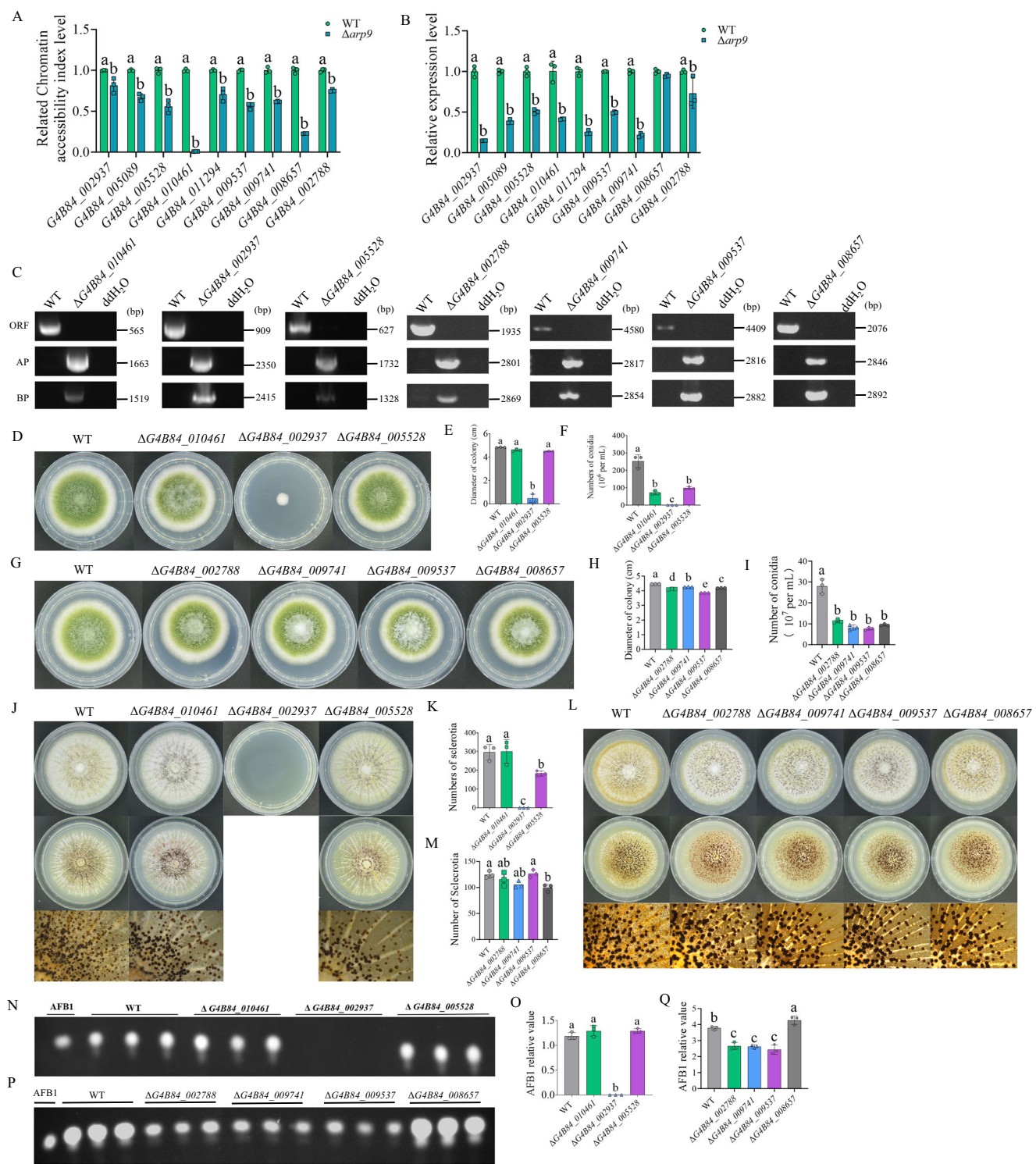

via ChART-qPCR, and the results demonstrated that Arp9 significantly promotes their chromatin accessibility (Fig. 8A), which is consistent with the ATAC-seq results. Further RT-qPCR analysis revealed that Arp9 enhances the expression levels of all the nine genes through chromatin remodeling (Fig. 8B). To explore the roles of M2DH, SMAP, PIK2, UMF1, CDR1, PSDP, and DAK (TKFC) in the regulatory pathway initiated by Arp9, the corresponding gene deletion mutants (ΔG4B84_010461, ΔG4B84_002937, ΔG4B84_005528, ΔG4B84_002788, ΔG4B84_009741, ΔG4B84_009537 and ΔG4B84_008657) were further constructed and confirmed by diagnostic PCR (Fig. 8C). The results revealed that all these mutants were involved in Arp9-mediated

regulation of colonial growth and conidiation, with SMAP showing the most prominent effect (Fig. 8D–I). The analysis of these mutants' roles in sclerotia formation demonstrated that the absence of PIK2, CDR1, and DAK (TKFC) significantly downregulated sclerotia formation (Fig. 8J, M). Notably, the absence of SMAP completely inhibited the germination and growth of fungal spores on the sclerotia-inducing medium (Fig. 8J, indicated by the empty space), which in turn totally blocked sclerotia formation. Furthermore, the AFB1 synthesis capacity of *A. flavus* was completely inhibited in the ΔG4B84_002937 strain (Fig. 8N, O) and significantly decreased in the ΔG4B84_002788, ΔG4B84_009741, and ΔG4B84_009537 strains (Fig. 8P, Q).

**Fig. 8 | The roles of the targets of Arp9 in fungal development and AFB1 biosynthesis. A** Chat-qPCR was performed to assess the relative CAI (Chromatin accessibility index) levels in the promoter regions of SMAP (G4B84_002937), PIKA (G4B84_005089), PIK2 (G4B84_005528), M2DH (G4B84_010461), PSD (G4B84_011294), UMF1 (G4B84_002788), CDR1 (G4B84_009741), PSDP (G4B84_009537) and DAK(TKFC) (G4B84_008657) genes in ΔArp9 and WT strains. **B** Relative expression levels of the above genes in ΔArp9 and WT strains. **C** The gene deletion mutants ΔG4B84_010461, ΔG4B84_002937, ΔG4B84_005528, ΔG4B84_002788, ΔG4B84_009741, ΔG4B84_009537 and ΔG4B84_008657 were verified by diagnostic PCR. **D** The spores of ΔG4B84_010461, ΔG4B84_002937 and ΔG4B84_005528 were incubated onto PDA under 37 °C for 4 d. **E** The histogram reflecting the colonial diameter of the above fungal strains according to the result of (**D**). **F** The spore number statistic result of the above fungal strains according to the result of (**D**). **G** The spores of ΔG4B84_002788, ΔG4B84_009741, ΔG4B84_009537, and ΔG4B84_008657 were incubated onto PDA under 37 °C for 4 d. **H** The histogram reflecting the colonial diameter of the above fungal strains according to the result of (**G**). (**I**) The spore number statistic result of the above fungal strains according to the result of (**G**). **J** The spores of ΔG4B84_010461, ΔG4B84_002937, and ΔG4B84_005528 were incubated onto CM media under 37 °C for 7 d (above panels); Middle panels were sprayed with 75% ethanol, and the lower panels were the enlarged pictures of the middle panels under a dissecting microscope. The empty position means no hyphal growth was observed. **K** The statistics of sclerotia number according to (**J**). **L** The spores of ΔG4B84_002788, ΔG4B84_009741, ΔG4B84_009537, and ΔG4B84_008657 were incubated onto CM media under 37 °C for 7 d (above panels); Middle panels were sprayed with 75% ethanol, and the lower panels were the enlarged pictures of the middle panels. **M** The statistical result of sclerotia number from (**L**). **N** The biosynthesis of AFB1 in WT, ΔG4B84_010461, ΔG4B84_002937, and ΔG4B84_005528 fungal strains was detected by TLC after being grown at 29 °C in dark for 7 d. **O** The relative biosynthesis levels of AFB1 from the ΔG4B84_010461, ΔG4B84_002937, and ΔG4B84_005528 strains were semi-quantified according to the results of the (**N**). **P** The biosynthesis of AFB1 in WT, ΔG4B84_002788, ΔG4B84_009741, ΔG4B84_009537, and ΔG4B84_008657 fungal strains was detected by TLC after being grown at 29 °C in dark for 7 d. **Q** The relative biosynthesis levels of AFB1 from the ΔG4B84_010461, ΔG4B84_002937, ΔG4B84_005528, ΔG4B84_002788, ΔG4B84_009741, ΔG4B84_009537, and ΔG4B84_008657 strains were semi-quantified according to the results of the (**P**). Data in (**A, B, E, F, H, I, K, M, P**) are presented as mean ± SD ($n = 3$). The unpaired two-tailed $t$-test was used to compare the statistical significance of (**A, B**). One-way ANOVA coupled with Tukey's multiple comparisons test was used in statistical significance analysis for (**E, F, H, I, K, M, P**). Bars in **A, B, E, F, H, I, K, M, P** with different top letters indicate significant differences ($P < 0.05$), whereas those with the same letter mean no significant differences. Source data are provided as a Source data file.

Collectively, the above results demonstrated that Arp9 regulated fungal morphogenesis and the biosynthesis of the secondary metabolite AFB1 through its target genes.

## Discussion

*A. flavus* is a worldwide distributed opportunistic pathogenic fungus that infects crops and animals, producing numerous highly toxic and carcinogenic aflatoxins, thereby seriously endangering human health[44]. To reduce or eliminate the threat posed by *A. flavus*, it is necessary to reveal the regulatory mechanisms underlying fungal pathogenicity and toxins synthesis. In particular, the biological function and underlying regulatory mechanism of histone H3K4 methylation−a key epigenetic modification−in *A. flavus* remain unclear. Our research group has previously focused on the regulatory mechanism of Set1, a subunit of the COMPASS complex, on H3K4 methylation[35]; however, the signaling pathway of the COMPASS complex remains to be further explored. Based on Bre2 (a key subunit of the COMPASS complex in *A.* flavus), we investigated how COMPASS chards the biosynthesis of fungal secondary metabolites and fungal virulence through chromatin remodeling via the CRF Arp9 under H3K4 methylation (Fig. 9).

Bre2 is a key regulatory factor in fungal virulence and AFB1 biosynthesis. It has been reported that deletion of *bre2* in *M. oryzae* can lead to serious defects in colonial growth and conidiation[45]. In *C. deneoformans*, Bre2 is involved in the activation of genes responsible for the yeast-to-hypha transition[46]. In this study, we found that *A. flavus* Bre2 plays a significant positive regulatory role in the morphogenesis of *A. flavus* (Fig. 1A−H). At the same time, we demonstrated that hyphal growth and sporulation on crop kernels inoculated with Δ*bre2* spores were dramatically reduced compared to those in the WT group (Fig. 2A−C), and Bre2 was a key regulator of the virulence of *A. flavus* against *G. mellonella* larvae (Fig. 2G−N). Furthermore, we found that Bre2 was involved in the positive regulation of the pathogenicity of *A. flavus* through upregulating the expression of secret protein effectors of fungal pathogens: CP1, Fgb1, LysM, MHP1, MPG1, Tox1, and WSC4 (the homolog of WSC3) (Supplementary Fig. 2A). For successful infection, fungal pathogens secret protein effectors to counteract the defense system of host. As a conserved secretory protein, CP1 belongs to SnodPort1 phytotoxin family. It has been reported to protect the wall of pathogenic fungi from enzymatic degradation; the expression level of *cp1* remains elevated during the infection process, and *cp1* knockout significantly attenuates the pathogenicity of *V. dahliae*[47].

Gene *fgb1* encodes a secretory fungal-specific β-glucan-binding lectin, which plays a key role in altering the composition and properties of the fungal cell wall and in suppressing the host immunity response triggered by β-glucan in the fungal pathogen *Piriformospora indica*[48]. And as a β-glucan-binding lectin, WSC3 enhances fungal resistance to cell wall stressors in *Pichia pastoris*[49]. In *Mycosphaerella graminicola*, LysM takes part in the protection of fungal hyphae against degradation by host hydrolytic enzymes and in the blockage of chitin-induced host defense responses. The colonization capacity of the Mg3LysM mutant was severely inhibited[50]. As one of hydrophobins, MHP1 shows highly induced expression during the colonization of *Magnaporthe grisea* on the host, and the absence of MHP1 reduces appressorium development in *M. grisea* and inhibits its infectious growth[51]. As a typical pathogenicity factor in *M. grisea*, MPG1 is expressed during appressorium formation and symptom development, and it is indispensable for infection-related development and the full pathogenicity of *M. grisea* on susceptible hosts[52]. As a necrotrophic effector, the absence of Tox1 from virulent *Stagonospora nodorum* could render it avirulent. The interaction of SnTox1 and host Snn1 leads to a series of host defense responses, including oxidative burst and pathogenesis-related gene expression[53]. The results suggested that Bre2 regulates fungal virulence through adjusting fungal morphogenesis and virulence-related protein effectors.

The Bre2 ortholog in *A. nidulans*, CclA, activates the expression of cryptic secondary metabolite clusters, thereby enhancing the production of monodictyphenone, emodin, and emodin derivatives[31]. As a notorious fungal secondary metabolite and the most abundant component in AFs, AFB1 produced by *A. flavus* is the most toxic mycotoxin[54]. In this study, we also focused on the biological effect of Bre2 on AFB1 biosynthesis in media and hosts and found that the AFB1 yield of Δ*bre2* mutant was significantly lower than that of the WT (Figs. 1I−L and 2D−F, N), indicating that Bre2 strongly regulated the biosynthesis of AFB1 in *A. flavus*. We found that AFs are non-negligible factors in Bre2-mediated fungal virulence toward animal hosts (Supplementary Fig. 2B, C). The aflatoxin gene cluster of *A. flavus* includes 30 genes, and which control AFB1 production through the polyketide pathway[55]. The activation of aflatoxin gene cluster is mainly regulated by pathway-specific transcription factors AflR and AflS[56]. The signaling pathway by which Bre2 regulates AFB1 biosynthesis was further explored by RT-qPCR, and the results indicated that Bre2 controls AFB1 biosynthesis through the aflatoxin gene cluster (Fig. 1L). In *A. nidulans*, the absence of the Bre2 ortholog, CclA, activates a series of secondary

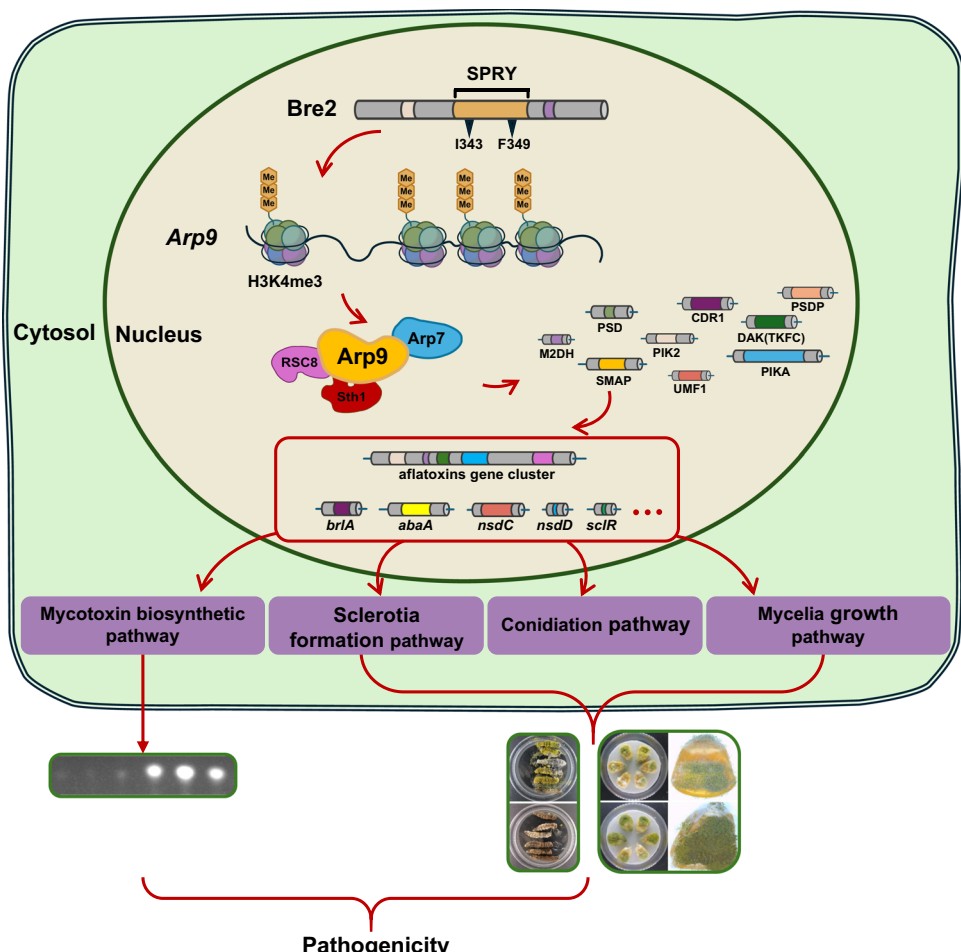

**Fig. 9 | Bre2 regulates fungal development and AFB1 biosynthesis through chromatin remodeling via Arp9.** Bre2 regulates mycotoxin biosynthesis, development, and virulence via its key target the CRF Arp9, through the trimethylation of H3K4. During the process, Arp9 form chromatin remodeling complex by interacting with RSC8, Arp7, and Sth1, by which Arp9 positively remodels the chromatin accessibility of SMAP, PSD, PIK2, M2DH, PIKA, DAK, UMF1, CDR1, and PSDP to regulate the canonical transcriptional factors of fungal development and secondary metabolism. Finally, through CRF Arp9 and its targets, Bre2 realized the epigenetic regulation of the virulence and secondary metabolites of *A. flavus*.

metabolites[31], while in *A. flavus*, the absence of Bre2 inhibits the yield of AFB1, which suggests that Bre2 undertakes different regulatory roles in different fungi or for different metabolites. To investigate whether the functionality of each subunit in the COMPASS complex is solely determined by complex integrity, the function of Sdc1 in this complex was abrogated by homologous recombination, and phenotypic analysis of the Δ*sdc1* strain and the complementary strain Com-*sdc1* showed that like Bre2, Sdc1 participates in fungal morphogenesis, but it does not participate in the regulation of AFB1 biosynthesis (Supplementary Fig. 8). The results demonstrated that the subunits in the COMPASS complex possess both the overall characteristics of the whole complex and subunit-specific properties in *A. flavus*. Furthermore, these results reflected that Bre2 regulates *A. flavus* virulence, including its development and the biosynthesis of secondary metabolites, through transcription factors in related signaling pathways.

Highly conserved Bre2 plays an important role in fungal histone methylation. Bioinformatics analysis showed that *A. flavus* Bre2 consists of 617 aa and contains a conserved SPRY_Ash2 domain. Then, we aligned Bre2 homologous proteins from other model species, including yeast, human, and other *Aspergillus* fungi. It was found that *A. flavus* Bre2 is more closely related to Bre2 from filamentous fungi. Furthermore, all the above orthologs contain a highly similar SPRY_Ash2 domain (Supplementary Fig. 1A, B), indicating that Bre2 is highly conserved during evolution[57]. The above results reflect that Bre2 might

play a similarly vital biological function from filamentous fungi to humans. Histone methylation was first reported in 1964 when Allfrey et al. discovered the methylation of arginine (R) and lysine (K) residues in the tails of histone H3, H4, and H2A[57]. As the only H3K4 histone methyltransferase in *S. cerevisiae*, the Set1/COMPASS complex is an excellent system for studying the relationship between H3K4 methylation and transcription[58]. The catalytic activity of Set1 needs to be maintained by the WRAD complex. As a member of the catalytic core of the COMPASS and WRAD, Bre2 is almost entirely responsible for the methylation of H3K4me1-3 in *Cryptococcus neoformans*[46]. In *S. cerevisiae*, the absence of Bre2 results in increased modification of H3K4me and H3K4me2 and a severe decrease in H3K4me3 at the 5′ end of active coding regions[59]. We found that Bre2 regulated trimethylation and dimethylation of H3K4 in a site-specific manner in *A. flavus* (Fig. 2Q).

Bre2 regulates fungal virulence through SPRY_Ash2 (SPRY). The SPRY domain, a core element of all MLL family histone methyltransferases, is highly conserved during evolution and is important for MLL methyltransferase activity by participating in protein-protein interactions[60–63]. The role of SPRY in the major biological functions of Bre2 was explored, and we found that, like the phenotype of Δ*bre2* mutant, SPRY and its key conserved amino acid residues (I343 and F349) play crucial roles in fungal morphogenesis, AFB1 biosynthesis, and fungal colonization on crop kernels (Fig. 3B–M). It has been reported that SPRY of Ash2L is one of the minimal requirements of

MLL[SET] activation[64]. When the conserved nonpolar amino acids Ile and Phe in SPRY were substituted with the polar amino acids Asn and Ser, the fungal phenotype tended towards that of the strains with complete deletion of the SPRY domain (Fig. 3), suggesting that the nonpolar nature of Ile and Phe plays a key role in the biological function of the SPRY domain[65,66]. Our results showed that Bre2 regulates the virulence of *A. flavus* through SPRY by mediating the methylation of H3K4me2-3, especially H3K4me3 (Fig. 3A). In human Ash2L, SPRY is responsible for its interaction with RbBP5 and DPY30[67], which suggests that in the *bre2*[ΔSPRY], *bre2*[I343N] and *bre2*[F349S] strains, the interaction of Bre2 with Swd1 and Sdc1 might be obstructed, and the disrupted COMPASS complex fails to perform its normal biological function properly. All the above results indicated that Bre2 regulates fungal growth, development, AFB1 biosynthesis, and the virulence of *A. flavus* through SPRY and its key conserved residues by maintaining the stability of the methyltransferase complex.

ChIP-seq analysis revealed the signaling pathway regulated by Bre2 through H3K4me3 modification. In *S. cerevisiae*, Bre2 is indispensable for the proper methylation of H3K4 and the activation of related genes[25,30]. In this study, an anti-H3K4me3 monoclonal antibody was used for ChIP-seq assay to screen for genes modified by Bre2 via the trimethylation of lysine residues. The results showed that the signal of H3K4me3-ChIP-seq reads was highly enriched in the 2 kb region upstream and downstream of TSS (Supplementary Fig. 5A), indicating that H3K4me3-modified chromatin fragments were mainly distributed in the promoter regions and transcription start sites. The above results are consistent with reports in other species[68,69], reflecting that the region regulated by Bre2-mediated H3K4me3 modification is conserved. A total of 2871 up-regulated DAPs were identified (Supplementary Fig. 5G), indicating that Bre2-mediated H3K4me3 exerts broad regulatory effects on the genome-wide expression of *A. flavus*. In *S. cerevisiae*, components of the COMPASS complex, including Bre2, repress the expression of maltose utilization genes during the late stages of fermentation[70]. In *A. nidulans*, Bre2 is associated with the production of monodictyphenone, emodin, emodin derivatives, and two polyketides[71]. Through ChIP-seq, we also found that Bre2 of *A. flavus* is involved in polyketide metabolism, the biosynthesis of other secondary metabolites, lipid metabolism, energy metabolism, carbohydrate metabolism, and amino acid metabolism (Supplementary Fig. 5I). The above analysis results showed that Bre2 regulates the genomic transcription of *A. flavus* through H3K4me3 modification, is widely involved in the adjustment of various primary and secondary metabolic pathways and participates in the modulation of various life processes of *A. flavus*, such as development, stress response and AFB1 production.

Arp9 is the key target of Bre2 in chromatin remodeling regulation. In fission yeast, the histone methyltransferase Clr4 can target pericentric repeats to establish heterochromatin[72]. H3K4 histone methyltransferase Set1 is essential for pervasive transcription and antisense-mediated gene silencing in yeast, especially for extensive chromatin remodeling at the promoters[73]. From in-depth analysis of ChIP-seq data, we found that the target genes regulated by Bre2-mediated H3K4me3 include a series of Snf2 family genes containing the DEAQ box helicase domain (AFLA_086690, AFLA_129070, AFLA_136160, and AFLA_136160). Numerous studies have shown that the chromatin remodeling family is a key determinant of epigenetic processes such as histone methylation[74–76]. The Snf2 family comprises a diverse set of enzymes with distinct biological functions, which are widely implicated in the regulation of critical biological processes such as stem cell differentiation and immune response by modulating chromatin structure[77–79]. As a representative member of this family, Snf2 has attracted growing attention. Studies have demonstrated that Snf2 plays a critical role in the growth of fungi such as yeast, and the related chromatin remodeling complexes are evolutionarily conserved from yeast to humans[76,80]. The Swi/Snf chromatin remodeling complex in *Candida albicans* can differentially regulate the expression of *MDR1* (a gene coding a drug efflux pump) and fluconazole resistance[81]. The INO80 chromatin-remodeling complex is essential for coordinating respiration, cell division, and periodic gene expression in *S. cerevisiae*[82]. ChIP-seq and qRT-PCR analysis revealed that Bre2 regulates the expression of *arp9*, an important member of Snf2 family, via H3K4me3 modification (Fig. 4A–C). Further ChIP-qPCR analysis revealed that Bre2 positively regulates *arp9* expression by trimethylation of H3K4 at the promoter and 5'-exon of *arp9* (Fig. 4D–H). Wagner et al. reported that the yeast chromatin-remodeling complex subunit Arp9 forms a stable ARP module with Arp7[83]. In *S. cerevisiae*, *arp9* mutants are either inviable or exhibit severe dysplasia[33]. In a drug-hypersensitive genetic background, overexpression of *arp9* conferred resistance to momilactone B in yeast[84]. Arp9 in *Penicillium oxalicum* plays essential roles in fungal development and the expression of the genes of cellulase and amylase[34]. Cairns et al. reported that Arp9 affects the growth of *S. cerevisiae*, and *arp9* mutants exhibit severely impaired growth[33]. In this study, we found absence of Arp9 caused defects in colony growth and sporulation (Supplementary Fig. 6D–F). In addition, we showed that upon the absence of Arp9, *A. flavus* failed to form sclerotia normally, indicating that Arp9 is involved in both asexual and sexual reproduction of *A. flavus*. We also found that Arp9 is involved in AFB1 biosynthesis in *A. flavus* by regulating the aflatoxin gene cluster (Supplementary Fig. 6M, N). The crop kernel model revealed that Arp9 is involved in sporulation and AFB1 biosynthesis on kernels (Supplementary Fig. 7A–C). These results suggested that Arp9 is one of the key downstream targets of Bre2 and demonstrated the critical role that Arp9 plays in the Bre2-triggered epigenetic signaling pathway regulating fungal virulence, proliferation, development, and secondary metabolite production in *A. flavus*.

ATAC-seq analysis revealed that Arp9 exerts its biological function by regulating downstream targets via chromatin remodeling. ATAC-seq analysis revealed that, compared with the WT strain, the Δ*arp9* strain exhibited 214 downregulated genes and only 22 upregulated genes, indicating that Arp9 remodels chromatin accessibility mainly by promoting chromatin opening, thereby enhancing the transcriptional activity of relevant genes (Fig. 7B). GO enrichment analysis of ATAC-seq data revealed that Arp9 is involved in metabolic processes, responses to stimuli, biological regulation, and signaling (Fig. 7E). KEGG pathway analysis showed that Arp9 is involved in regulating a variety of important metabolic pathways by promoting chromatin opening, including primary metabolism and secondary metabolism (Fig. 7F). M2DH participates in D-mannitol oxidation to facilitate the production and mobilization of mannitol, thereby contributing to *A. fumigatus* resistance against host defense strategies[85]. SMAP is a membrane protein that may play a role in stroma-supported erythropoiesis and erythropoietic activity in mice[86]. *S. cerevisiae* harbors two genes that encode phosphatidylinositol 4-kinases, namely STT4 and PIK1 (PIKA). STT4 is required for the maintenance of vacuole morphology, cell wall integrity, and actin cytoskeleton organization, and it has been found that the STT4 mutant strain is sensitive to staurosporine and presents an osmoremedial phenotype. In contrast, PIK1 is essential for normal secretion, the Golgi and vacuole membrane dynamics, and endocytosis[86,87]. PSD plays a crucial role in mycelial growth, sexual and asexual reproduction, virulence, lipid droplet formation, and autophagy of *F. graminearum*, and is essential for cell wall integrity and virulence in *C. albicans*[88,89]. UMF1 (MFS transporter) is involved in the upregulation of drug resistance in the pathogen *Acinetobacter baumannii* to cefiderocol[90]. The multidrug transporter CDR1 is one of the key factors in the development of azoles resistance in *Candida albicans*[91]. PSDP is involved in neutrophil apoptosis in ulcerative colitis-associated carcinogenesis via Rac1-dependent immune responses[92]. And DAK (TKFC) plays an important role in the development of HCC[93]. Our CHART-PCR and RT-qPCR analysis revealed that Arp9 modulates morphogenesis, virulence, and AFB1

biosynthesis in *A. flavus* by enhancing the expression of related genes (including M2DH, SMAP, PI4K2, PI4KA, PSD, UMF1, CDR1, PSDP, and DAK (TKFC)) via chromatin remodeling (Fig. 8A, B). We further explored the roles of M2DH, SMAP, PI4K2, UMF1, CDR1, PSDP, and DAK (TKFC) in the Arp9-initiated regulatory pathway using gene knockout experiments. Our results revealed that these factors are key targets of Arp9 in regulating colony growth, development, and AFB1 biosynthesis in *A. flavus*, with SMAP being particularly critical (Fig. 8D–Q). Our RT-qPCR analysis demonstrated that SMAP regulates sporulation, sclerotia formation, and AFB1 biosynthesis via global regulators and pathway-specific transcription factors (Supplementary Fig. 9). The above results revealed that Arp9 is the key target of Bre2 in regulating secondary metabolism and virulence in *A. flavus* by enhancing the chromatin accessibility of downstream genes.

In conclusion, we elucidated the regulatory mechanism of H3K4 methylation mediated by Bre2 on the growth, development, AFB1 biosynthesis, and virulence of *A. flavus*. We also revealed the epigenetic modulation mechanism triggered by the Bre2-Arp9 axis, which regulates the target genes by initiating chromatin remodeling (Fig. 9). This study lays a foundation for the early biological prevention and control of *A. flavus* and aflatoxin contamination, and provides potential targets for the clinical treatment of fungal infections and the development of antifungal drugs.

## Methods

### Strains and cultural conditions
*A. flavus* Δ*ku70* Δ*pyrG* was used as the parental strain for constructing fungal mutants. All fungal strains were constructed via homologous recombination, and all strains used for the analyses are listed in Supplementary Data 2. All primers were synthesized by Fuzhou Shangya Biotechnology Co., Ltd. and are listed in Supplementary Data 3–5. Potato dextrose agar (PDA, 39 g/L, BD, Difco, Franklin, NJ, USA) was used for the assessment of mycelial growth and sporulation, while complete medium (CM, 6 g/L tryptone, 6 g/L yeast extract, 10 g/L glucose) was employed for sclerotia production assay. Potato dextrose broth (PDB, 24 g/L, BD, Difco, Franklin, NJ, USA) was utilized for mycotoxin production analysis. Uridine and uracil were supplemented as required to complement the auxotrophic marker (*pyrG-*)[9]. All experiments were performed in triplicate.

### Bioinformatics analysis
Bre2 homologs from 16 fungal species (including: *A. flavus*, *A. tamarii*, *A. nomius*, *A. bombycis*, *A. bertholletius*, *A. tanneri*, *A. steynii*, *A. glaucus*, *A. clavatus*, *A. fumigatus*, *A. violaceofuscus*, *A. piperis*, *A niger*, *A phoenicis*, *S. cerevisiae*, and *Homo sapiens*) were downloaded from the NCBI BLAST (http://www.ncbi.nlm.nih.gov). The amino acid sequences of Bre2 orthologs from all 16 selected species were analyzed by MEME. A phylogenetic tree of the Bre2 proteins from all these 16 species was further established with the software MEGA11.0 (MEGA11: Molecular Evolutionary Genetics Analysis Version 11). Protein domains of these species were analyzed using the software SMART, and further domain visualization of these proteins was completed by DOG2.0[9]. The homologs of Arp9 were also downloaded from the NCBI BLAST and were analyzed following the methods mentioned above.

### Construction of mutant strains
All mutant strains were constructed following the protocol of homologous recombination[9]. Specifically, the *bre2* deletion strain (Δ*bre2*) and the complementary strain (Com-*bre2*) were constructed using the same approach. For the preparation of Δ*bre2* strain, the 5′ flanking region (homologous arm, 5′FR), 3′ flanking region (homologous arm, 3′FR), and *pyrG* gene of *A. fumigatus* was amplified, and the 5′FR-*pyrG*-3′FR fusion fragment was obtained by in vitro fusion PCR. The fused PCR product was introduced into CA14 protoplasts via polyethylene glycol-mediated transformation[94], and the candidate transformants

were screened by incubation in uridine/uracil-free resuscitation medium (372.4 g/L sucrose, 3 g/L NaNO$_3$, 1 g/L K$_2$HPO$_4$, 0.5 g/L KCl, 0.5 g/L MgSO$_4$, 0.01 g/L FeSO$_4$, 10 mmol/L ammonium tartrate, 0.5% agar) for 3–5 days, which were then confirmed by diagnostic-PCR and Southern blotting analysis. The construction of the Com-*bre2* strain was also performed following the protocol of homologous recombination[95]. The candidate Com-*bre2* strains were verified by diagnostic PCR and qRT-PCR. The domain deletion fungal strain *bre2*$^{\Delta SPRY}$, point-mutation strains *bre2*$^{I343N}$ and *bre2*$^{F349S}$, as well as gene deletion stain Δ*arp9* and complementary strain Com-*arp9* were constructed and verified vis the same method. The construction of the Bre2-mCherry strain and the Bre2-eGFP strain were performed following the same homologous recombination method[96]. HA/Strep dual-tagged strains were also prepared via homologous recombination. Briefly, the 3HA-*pyrG* gene fragment was ligated to the 3′ end of the *Arp9* gene by fusion PCR. Then, an Apr9-3HA single-tagged strain was obtained by transforming CA14 protoplasts with fused the fused *Apr9*-3HA-*pyrG* fragment. After deleting the *pyrG* gene from the Apr9-3HA single-tagged strain under the stress of 2 mg/mL 5-FOA (5-fluoroorotic acid), the *RSC8*-Strep-*pyrG* fragment was introduced into the protoplasts of this single-tagged strain via homologous recombination, and the Apr9-3HA/RSC8-Strep dual-tagged strain was screened on uracil/uridine-free resuscitation medium. Similarly, the Apr9-3HA/Sth1-Strep and Arp9-3HA/Arp7-Strep dual-tagged strains were constructed using the same method. Subsequently, all dual-tagged *A. flavus* strains were verified by sequencing (Beijing Tsingke Biotech Co., Ltd.) and Western blotting. All primers are listed in Supplementary Data 3.

### Quantitative RT-PCR (RT-qPCR) analysis
The expression levels of target genes were determined by RT-qPCR[97]. Total RNA was extracted using TRIzol reagent (Vazyme Biotech, Nanjing, China), which was then reverse-transcribed into cDNA using the First-Strand cDNA Synthesis Kit (Transgen biotech, Beijing, China). RT-qPCR was performed using the QuantStudio 1 plus PCR system (Applied Biosystems, MA, USA). Primers used for RT-qPCR are listed in Supplementary Data 4. *β-tubulin* was used as the inner reference gene. All experiments were repeated in triplicate.

### Phenotypic and aflatoxin analysis
The spores ($10^4$ conidia/mL) of each fungal strain were inoculated onto PDA medium for sporulation and onto CM medium for sclerotium formation[96]. For sporulation analysis, the *Petri* dishes were kept in the dark incubator at 37 °C for 5 days; conidia were then collected, diluted in 3 mL of 0.05% Tween-20, and finally counted using a hemacytometer. For sclerotium formation analysis, the *Petri dishes* were sprayed with 75% ethanol to remove conidia and mycelium after 7 days of incubation, and sclerotia were counted and photographed using a Leica MZ75 dissecting microscope coupled with a Leica DC50LP camera (Leica Microsystems Inc., Buffalo Grove, IL, USA). Aflatoxin was extracted with chloroform[98]. Thin-layer chromatography (TLC) and high-performance liquid chromatography (HPLC) quantification were also carried out to measure aflatoxin production by the above fungal strains[97]. Spores ($10^7$ conidia/mL) of the indicated fungal strains were inoculated into 10 mL PDB at 29 °C for 7 days. A 4-mL aliquot of the culture was mixed with an equal volume of dichloromethane. The lower layer (3 mL) was aspirated and air-dried. The dried aflatoxin was then redissolved in 100 µL dichloromethane, and 10 µL of the solution was analyzed by TLC. Toxin bands were visualized using a UV light imaging system. For HPLC analysis, the aflatoxin extract was redissolved in methanol and filtered (0.22 µm). The HPLC parameters were as follows: SunFire™ C$_{18}$ column (Waters, Milford, Ma, USA); mobile phase, water:methanol:acetonitrile (56:22:22, v/v/v); column temperature, 42 °C; injection volume, 20 µL per sample; elution at a flow rate of 1.0 mL/min for 15 min. AFs were detected using a fluorescence detector (Waters 2475 Multi λ Fluorescence Detector, USA). Crop

kernels, including corn and peanuts, were selected to assess plant infection by *A. flavus*[99]. To explore the role of Bre2 and Arp9 in the infectivity of *A. flavus* to animals, *Galleria mellonella* and *Bombyx mori* larvae were chosen as hosts in infection assays[100–102]. *G. mellonella* larvae (1 cm) were purchased from Jiyuan City (Henan, China), and *B. mori* larvae (2nd instar) were purchased from Changzhou City (Jiangsu, China).

## Western blotting analysis

Histone methylation sites and levels regulated by Bre2, as well as the interaction proteins of Arp9 were determined by Western blotting[96]. Spores ($10^4$ conidia/mL) of fungal strains, including WT, Δ*bre2*, Com-*bre2*, *bre2*$^{\Delta SPRY}$, *bre2*$^{I343N}$, and *bre2*$^{F349S}$ were inoculated in PDB medium and cultured at 29 °C for 48 h. The membrane was incubated with 1:1000 diluted primary rabbit monoclonal antibodies at 4 °C overnight. The primary antibodies included H3K4me (ab176877; Abcam), H3K4me2 (ab32356; Abcam), H3K4me3 (ab213224; Abcam), H3K9me (ab176880; Abcam), H3K9me2 (ab176882; Abcam), H3K9me3 (ab176916; Abcam), H3K36me (ab176920; Abcam), H3K36me2 (ab176921; Abcam), H3K36me3 (ab282572; Abcam), H3 (ab1791; Abcam), anti-HA tag antibody (ab236632; Abcam), and anti-Strep-tag II antibody (ab307676; Abcam). Subsequently, the membranes were incubated within a 1:5000 dilution of goat anti-rabbit IgG antibody (H&L (bs-0295G-HRP) Bioss) at room temperature for 1 h. Finally, the membrane was developed using equal volumes of Immobilon™ Western HRP substrate luminol reagent and Immobilon™ Western HRP substrate peroxide solution, and imaged using a G:Box XT4 chemiluminescence and fluorescence imaging system (Shanghai Gene Co., Ltd., China)[35].

## ChIP-seq

Genes directly regulated by Bre2 were screened via chromatin immunoprecipitation (ChIP)[103]. For ChIP experiments, WT and Δ*bre2* fungal strains ($10^4$ spores/mL) were grown in YES liquid medium at 180 rpm for 24 h. Subsequently, cross-linking, DNA sonication, and chromatin immunoprecipitation were performed[104]. After sonication, the remaining chromatin solution was divided into two parts: one was incubated with the addition of 10 μg of the antibodies (anti-H3K4me3), and the other was incubated without antibodies (mock). Immunoprecipitated DNA was used for sequencing, and the sequencing library was prepared with Illumina-sequenced NEXTFLEX ChIP-Seq Libraries Kit (NOVA-514120, Bioo Scientific) followed the PE 150 protocol provided by the kit. DNA sequencing was carried out by Wuhan IGENEBOOK Biotechnology Co., Ltd.

## ATAC-seq

Genes regulated by Arp9 were identified via ATAC-seq[105]. WT and Δ*arp9* strains were inoculated into PDB at 29 °C, 180 rpm, for 3 days, after which the hyphae were frozen with liquid nitrogen. Samples were lysed in lysis buffer (10 mM Tris-HCl (pH 7.4), 10 mM NaCl, 3 mM MgCl$_2$, and 0.1% NP-40) for 10 min to isolate nuclei. Then, the nuclei were incubated with the Tn5 transposome and tagmentation buffer at 37°C for 30 min (Vazyme Biotech). Libraries were prepared by PCR, assessed with a Bioanalyzer and Q-bit, and sequenced using an Illumina NovaSeq/DNBSEQ-T7 platform with 150-bp paired-ends reads by iGeneBook Biotechnology Co., Ltd. (Wuhan, China). Data analysis was performed according to the protocols for ATAC-seq data processing[106]. Raw sequence reads were initially processed for quality control using FastQC (version: 0.11.5), and adapter sequences and poor-quality reads were removed using Trim-momatic (version 0.36)[107]. Subsequently, the remaining reads were mapped to the reference genome of *A. flavus* strain NRRL 3357 using hisat2 (version: 2.0.1-beta)[108]. Pairwise Spearman correlations between any pair of ATAC-seq samples were calculated. MACS2 (version 2.1.0) was used to call peaks, and motifs were predicted using HOMER Software (version

3). The criteria for screening differential peaks were $P < 0.05$ and Fold<0.

## Chromatin accessibility real-time PCR (CHART-PCR)

The regulation of target gene chromatin accessibility by Arp9-mediated chromatin remodeling was confirmed by CHART-PCR assay[109]. The mycelium was ground into powder under liquid nitrogen. 0.05 g the powder was suspended in 500 μL nuclease digestion buffer (250 mM sucrose, 60 mM KCl, 15 mM NaCl, 0.05 mM CaCl$_2$, 3 mM MgCl$_2$, 0.5 mM DTT, 15 mM Tris−HCl pH7.5). 50 μL DNaseI was added, and the digestion reaction was carried out at 25 °C for 2.5 min. The reaction was stopped by adding 1 volume of 40 mM EDTA and 2% SDS. This mixture was extracted twice with phenol-dichloromethane and once with dichloromethane, then treated with 10 μg/mL RNase A at 37 °C for 15 min. DNase I specifically recognizes DNase I-hypersensitive sites, which are transcriptionally activated chromatin regions, and catalyzes digestion. Due to this characteristic, regions susceptible to DNase I digestion correspond to open chromatin. qPCR was performed on DNaseI-treated samples to measure the relative abundance (accessibility) of target regions. Results were calculated using the $2^{-\Delta\Delta Ct}$ method, normalized to the housekeeping gene tubulin, and expressed as the relative chromatin accessibility index (CAI). Primer sequences are listed in Supplementary Data 5.

## Immunoprecipitation and mass spectrometry

The interacting proteins of Arp9 were identified via immunoprecipitation and mass spectrometry[110]. Arp9-3HA samples were ground in liquid nitrogen, and every 100 mg powder was suspended in 1 mL IP lysate and 10 mL PSFM at 4 °C for 1 h using the Pierce™ Classic Beads IP/Co-IP Kit (Thermo SCIENTIFIC™, 88804). The supernatant was collected and incubated with HA Pierce™ Beads at 4 °C for 1-3 h; the beads were then collected and washed with cold immunoprecipitation wash buffer. Immunoprecipitated proteins were resolved by SDS-PAGE and visualized by silver staining and Western blotting. For mass spectrometry analysis, concentrated samples were separated by SDS-PAGE, and the remaining samples were sent to Applied Protein Technology Biotechnology Co., Ltd. (Shanghai, China) for mass spectrometry detection and analysis.

## Statistics and reproducibility

All results are presented as the means of triplicate, with error bars representing the standard deviation (SD). Statistical analysis was performed using GraphPad Prism 10 (La Jolla, CA, USA). One-way ANOVA was used to evaluate overall differences among groups for a single independent variable, and Tukey's multiple comparisons test was subsequently applied to determine statistical significance between specific groups. An unpaired two-tailed *t*-test was used to compare differences between two groups. The log-rank test was used to compare differences in median survival time among animal injection groups. Statistical significance was defined as $P < 0.05$.

## Reporting summary

Further information on research design is available in the Nature Portfolio Reporting Summary linked to this article.

# Data availability

All primers were synthesized by Fuzhou Shangya Biotechnology Co., Ltd. and are listed in Supplementary Data 3–5. ChIP-seq and ATAC-seq data have been deposited in NCBI's Gene Expression Omnibus (GEO) under accession GSE285749 and GSE285953. The proteomic data from mass spectrometry analysis have been submitted to the ProteomeXchange Consortium via the PRIDE partner repository with the dataset identifier PXD060646. Source data are provided with this paper.

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

## Acknowledgements

This work was generously supported by the grants of the National Natural Science Foundation of China (Grant No. 32470207 awarded to Z.Z., Grant No. 32070140 awarded to Z.Z.) and the Nature Science Foundation of Fujian Province (Grant No. 2021J02026 awarded to Z.Z.). We especially thank Professor Jun Yuan, Xiuna Wang, Yu Wang, and Xinyi Nie for their support in instrument maintenance and reagent ordering.

## Author contributions

Conceptualization: Z.Z. and S.W.; Investigation: X.P., M.S., D.W., D.M., H.L., L.C., and Y.L.; Writing and original draft: Z.Z., L.C., and M.S.; Writing, review and editing: Z.Z., S.W., L.C., and M.S.; Validation: Z.Z., M.S., and S.W.; Formal analysis: Z.Z., L.C., S.W., M.S., and X.M.; Supervision: S.W and Z.Z.; Funding acquisition: Z.Z.

## Competing interests

The authors declare no competing interests.
