## [Peer Review file · Nature Communications]

The COMPASS-subunit Bre2 targets CRF Arp9 to regulate AFB1 anabolism and virulence in *Aspergillus flavus*

Corresponding Author: Professor Shihua Wang

Version 0:

Reviewer comments:

Reviewer #1

(Remarks to the Author)

In this study, the authors first explore the consequences of inactivating the COMPASS subunit Bre2 in different biological processes of *A. flavus*. They show that Bre2 deletion ($\Delta bre2$) reduces the expression of key transcription factors and thereby affects fungal morphogenesis and the level of the secondary metabolite AFB1. In agreement with these results, they show that Bre2 is required for virulence *A. flavus* to silkworm. They further show that *A. flavus* is localized in the nucleus and that deletion or mutations in the SPRY phenocopy the $\Delta bre2$ deletion mutant. Mapping the peaks of H3K4me3 in WT reveals that this mark is mainly located in promoters and exons of genes notably involved in metabolite synthesis and in several genes of the SNF2 family. Interestingly, they show that Bre2 and Arp9 are each required for the expression of the one by the other. In the second part of the manuscript, the authors show that the deletion of Arp9 produces effects similar to the deletion of Bre2 indicating that Arp9 collaborates with Bre2 to modulate fungal morphogenesis and metabolite production. ATAC-seq experiments reveal that Arp9 acts by remodeling the promoter of target genes, including genes involved secondary metabolites related process. In the paper, it is improper to write (lane 298): "H3K4me3 modification catalyzed by Bre2"; H3K4me3 is catalyzed by the Set1 complex

The experiments are solid and support the conclusions of this manuscript. My major problem is with the title, the summary, and the role is assigned to Bre2. Actually, the observed effects are not specifically due to Bre2, but rather to the absence of H3K4me3 and the reduction of H3K4me2. It is likely that the deletion of Sdc1 would produce the same effects as that of Bre2. The more appropriate title would be "H3K4me3 promotes CRF Arp9 expression to regulate the bio synthesis of secondary metabolite AFB1 and virulence in *Aspergillus flavus*". In this sense, part of the novelty of the article has already been revealed in their previous publication: "Methyltransferase AfSet1 Is Involved in Fungal Morphogenesis, AFB1 Biosynthesis, and Virulence of *Aspergillus flavus*".

The results on Arp9 and the link between H3K4me3 and the RSC complex are more novel and echo the MS of Schlichter and Cairns BR (PMID: 15775977).

Specifically, on the results, I have few comments:

- In figure 1K, what are the neighboring peaks of the AFB1 peak which also disappear in the $\Delta bre2$ mutant? Specify the abbreviation TLC
- H3K4me3 ChIP-seq reveal 2871 enriched genes and 723 depleted genes (WT compared to the $\Delta bre2$ strain). What are these 723 depleted genes ?
- Please provide the 63 proteins pulled down by Arp9.

Reviewer #2

(Remarks to the Author)

The manuscript describes regulation of a target to control the biosynthesis of secondary metabolites and virulence in *Aspergillus*. I appreciate the authors' robust process used for genetic confirmations. The study is well presented and organized.

Grammatical clarity could be improved.

Define COMPASS and ATAC

Add reference to line 88 (i.e., Our group previously reported...)

Line 125: italics to 'bre2'

Line 130: CM defined already?

Line 136: TLC and HPLC defined? Check throughout if terms from methods are defined within the Results for first use

Figure 1 – suggest increasing the size of the plots A-L (excluding K) to improve readability and decrease empty white space on the figures.

Most figures appear too small within the panels – sizes could be adjusted to optimize the figure size and reduce white space.

Reviewer #3

(Remarks to the Author)

In the manuscript entitled "The COMPASS subunit Bre2 targets CRF Arp9 to regulate the bio synthesis of secondary 1 metabolite AFB1 and virulence in *Aspergillus flavus*", the authors focused on the gene of Bre2 subunit of the histone methyltransferase complex and showed that bre2 is involved in H3K4 trimethylation in *A. flavus*. Through ChIP-seq analysis, they found that the gene of Arp9, which is involved in chromatin remodeling, is located downstream of bre2. The analysis of bre2 and arp9 knockout strains demonstrated that these genes are essential for the morphological development and aflatoxin production of *A. flavus*. Additionally, for bre2, they revealed functionally important protein domains and amino acid residues. Using epigenetic analysis of arp9 mutants, they investigated downstream genes affected by Arp9-mediated chromatin remodeling, constructed knockout strains for some of these genes, and examined their phenotypes.

The authors utilized a variety of approaches including seq analysis on the generated mutant strains to elucidate the relationship between epigenetic modifications—initiated by histone H3K4 trimethylation—and aflatoxin production of *A. flavus*, the notorious mycotoxigenic fungus. The experimental results clearly establish that arp9 functions downstream of bre2 and suggest that Arp9 exerts a distal effect on the aflatoxin biosynthetic pathway and morphological development pathway through SMAP and other factors.

However, the manuscript does not fully elucidate the mechanism by which Bre2 and Arp9 regulates aflatoxin production and morphological development. Although the ATAC-seq analysis identified several genes as potential components of the Arp9 regulatory pathway, only SMAP knockout strains exhibited significant effects on aflatoxin production and morphology, leaving the relationship between SMAP and these factors unclear. Simply to say, in Figure S6, the arrow which leads to the red boxes enclosing the gene clusters is unidentified. Due to this limitation, the overall framework linking histone H3K4 trimethylation initiated by Bre2 to aflatoxin production has not been fully clarified. Further analyses are needed to bridge this missing link.

The authors concluded that Bre2 is related to the colonization of peanuts and maize based on experiments using the poorly growing deletion strain Δ bre2. However, since poor growth is associated with colonization, it is not possible to discuss a direct relationship between Bre2 and colonization ability. Similarly, in the infection experiment with silkworms, the strong relationship between the poor growth of the deletion strain and its virulence makes it difficult to discuss the relationship between Bre2 and virulence in silkworms. Additionally, it should be noticed that the relationship between fungal AF productivity and pathogenicity to silkworms is unclear. In fact, the Δ bre2 strain still maintains lethality, with more than half of the silkworms dying within 6 days. As the same methods were used in experiments using other poorly growing deletion strains, no conclusions can be drawn from this paper regarding the relationship between Bre2 and virulence in *A. flavus*.

Specific comments:

The structure and explanations of the figures are inadequate, making it difficult for readers to understand the intended information. This issue is particularly pronounced from Figure 5 onward. Specifically:

- Figure 4A: Appears to be a screenshot from IGV, but lacks sufficient explanation.
- Figure 5B: The meaning of the mass spectrum is unclear.
- Figure 5C-E: The differences between lanes are not explained, and it is difficult to read what the detected bands represent.
- Figure 5F-H: Additional explanations for the bands are needed.
- Figure 6A: The descriptions of the lanes and labels lack.
- Figure 7: The text is too small to read. The labeling may be inappropriate (e.g., "arp9" should be " Δ arp9", isn't it?). The same issue applies to Figure S3.
- Figure 8A: The method for calculating the CAI index is unclear. Is there an easier metric to understand?

In terms of overall structure of the manuscript, the Results section includes elements that belong in the Discussion, while the Discussion section reads more like a Results and Discussion section, mixing result descriptions with interpretations. A thorough revision is needed to clearly separate these sections.

Line 40-42: As relationship of AF productivity of *A. flavus* with its pathogenicity in humans has not been proved, this sentence should be separated to avoid misunderstandings.

Figure 11: Spots are overlapped. It necessary to show the AFB1 spot clearly.

Figure 2H: It's necessary to show the data of a week.

Figure 2K: Rf values are different among the spots.

Line 215: The website should be included in the Methods section.

Figure 3A: The intensity of the H3 bands varies greatly between lanes, making it unsuitable as a control.

Line 313-314: The statement "the enrichment level in WT 313 was significantly higher (more than twice times)" lacks numerical data, making it unclear.

Line 363: The sentence "Immunoprecipitation was used to enrich ~" is unclear because Figure 5A presents a Western blot result, and it is not evident what was enriched.

Line 364: Information about the outsourced company should be placed in the Methods section.

Lines 365-367: There is an inconsistency between Line 365, which states "totally 63 proteins," and Line 367, which states "64 proteins." Additionally, it would be helpful to provide a table listing these proteins including the five screened-out proteins.

Line 370: What is the reasoning for stating that "~was found to tightly interact with Arp9"?

Lines 374-375: It is unclear how the dual-tagged *A. flavus* was constructed and verified. In addition, the company name should be included in the Methods section.

Line 408: Does this statement align with the observation that *bre2* gene expression was reduced in the *arp9* knockout strain?

Lines 451-452: Explanation is needed for the criteria for selecting target genes in ATAC-seq analysis. Were they screened based on GO terms?

Line 456: As mentioned above, the chromatin accessibility index is not well defined. The Methods section only mentions that it was calculated using the $2^{-\Delta\Delta CT}$ method, which is typically used for qPCR quantification.

Reviewer #4

(Remarks to the Author)

Version 1:

Reviewer comments:

Reviewer #1

(Remarks to the Author)

The authors did not respond to my major criticism regarding the specific role that Bre2 could play in relation to another subunit of the Se1 complex, such as Scd1. To validate a number of the article's conclusions, it is essential to compare and show in the article the differences in phenotypes between *bre2* Δ and *scd1* Δ .

The new title is incomprehensible.

Reviewer #2

(Remarks to the Author)

The authors have addressed my points; I appreciate the grammatical clarity and figure revisions.

Reviewer #3

(Remarks to the Author)

The manuscript has been substantially improved by the addition of new experiments, revisions of the text, and significant improvements to the figures (including enlargement of labels). These efforts have strengthened the work overall. However, several major issues remain that should be addressed.

Major comments:

1. Four additional genes were proposed as putative targets of Arp9, and corresponding knockout mutants were generated and analyzed. However, aside from Stromal membrane-associated protein (SMAP, G4B84_002937), none of the knockouts appear to have a substantial impact on aflatoxin production.

The role of SMAP in aflatoxin biosynthesis, fungal growth, and conidiation has not been previously reported and remains unclear. The mechanism underlying the observed effect of SMAP deletion is entirely unknown. In the Discussion, the authors only mention its involvement in stroma-supported erythropoiesis and erythropoietic activity in mice, which has no apparent relevance to *Aspergillus flavus*. The authors should therefore provide at minimum a plausible mechanistic hypothesis, and ideally, supporting validation experiments.

2. In the Materials and Methods, the Data Analysis section initially states that "All data were expressed as mean \pm standard

deviation,” but later specifies that “Error bars represented the standard error of at least three repetitions.” Which is correct? Furthermore, the section indicates that statistical analysis was performed using Tukey’s test. However, Tukey’s test is not appropriate in all cases (e.g., for comparisons between two groups). It is also unclear whether proper attention was paid to the issue of multiple comparisons. Statistical methods should be described not only in Materials and Methods but also clarified in each relevant figure legend where statistical data are presented.

3. It is not appropriate to directly compare the virulence of the WT strain and the Δ bre2 strain when there is a significant difference in their growth. The authors argue that the Δ bre2 strain exhibits reduced virulence because it has a weaker impact on the immune system of the larvae compared to the WT strain. However, this reduced impact may simply be due to the impaired growth of the Δ bre2 strain, rather than a direct consequence of the Bre2 deletion. Additionally, the authors suggest that the survival rate of larvae injected with extracts from the WT and Δ bre2 strains is related to the amount of AF contained in those extracts. To support this claim, it is necessary to conduct experiments using authentic AF at varying concentrations, and to quantify the AF content in the extracts before repeating the injection experiments.

4. There remain numerous grammatical errors, especially in the newly added texts with purple highlight (e.g., incorrect verb tenses). The manuscript should undergo professional English editing.

Minor comments:

1. Title: The expression “by H3K4me3” is unclear. In addition, abbreviations such as “me3” should be avoided in the title.
2. It would be appropriate to mention SMAP in the summary.

3. Line 47: The primary cause of aspergillosis is *A. fumigatus* rather than *A. flavus*. The relevance of the present study to aspergillosis is uncertain, and therefore this description should be deleted.

4. Line 334: In “the enrichment level of this chromatin fragment from WT is 21.1 times ($P < 0.05$) of that from Δ bre2 strain,” it is unclear how the enrichment level was calculated, and thus the meaning of “21.1 times” is ambiguous. Moreover, without information on sample size and the specific statistical test applied, the interpretation of reported P value is uncertain.

5. Line 350 (Figure 4): The label “H3K36me3” appears to be a mistake; it should likely read “H3K4me3.”

6. Lines 385–386 and Figure 5B: The name of the company should be reported in the Methods section, while the description here should focus on clarifying what experiments were performed and how the data were analyzed. In particular, Fig. 5B raises questions: based on the MS peaks of Arp9-3HA and WT, it seems unlikely that Arp9-3HA contains peaks corresponding to 63 additional proteins compared with WT. Were these peaks derived from peptides following tryptic digestion, or from another protease?

7. Line 390: The criteria by which the five proteins were “screened out” should be clearly defined.

8. Lines 397–414: This section contains many grammatical errors and requires thorough editing. A concise explanation of the “Strep” tag is also needed.

9. Figure 5 (F–H): Each panel requires a more detailed explanation. What detection methods were used—silver staining or Western blotting (and if Western blotting, with which antibody)? What proteins do the detected bands represent? In panel H, the band sizes are not indicated.

10. Figure 6A: The meaning of “AP” and “BP” should be explained.

11. Lines 473–476: From Fig. 7F, it does not appear that “metabolism, oxidative phosphorylation, and fatty acid degradation” are particularly heavily enriched. Rather, a variety of KEGG categories seem to be represented. What is the basis for concluding that these specific categories are heavily enriched?

12. Line 478: The statement “especially oxidative and secondary metabolites related process” does not seem well supported by the information in Fig. 7. A more detailed explanation of results of the KEGG and GO term enrichment is required.

13. Line 490: The basis for narrowing down 214 downregulated genes from the ATAC-seq results to nine candidate genes should be described. From which GO terms were they selected? Were there other genes belonging to the same GO terms that were excluded?

14. Line 511: “Fig. 8M” likely refers to “Fig. 8O.” Additionally, the phrase “significantly decreased” should be reserved for statistically significant differences. Although the AFB1 spot appears weaker, the criteria for significance are not provided.

15. Figures 8J and 8L: Each panel shows two plate images (top and bottom), but this is not described in the figure legends. Furthermore, in panel L, there should be no “empty position,” yet the legend mentions.

16. Discussion: I suggest that the Discussion be made more concise, focusing on the insights derived from the findings rather than merely summarizing the results.

17. Figure S7: This figure is important for understanding the manuscript, and I recommend that it be moved from the Supplementary section to the main figures (e.g., Figure 9). However, several points require clarification. Currently, it appears that all nine genes affected by Arp9, including SMAP, influence the aflatoxin gene cluster and other genes (red boxes). In reality, their effects are likely to differ. For example, gene G4B84_008657 (DAK) clearly does not affect aflatoxin production. It would be preferable to draw arrows only from the strongly influential genes, such as SMAP. In addition, the term “other biosynthetic gene clusters” is vague and may be better omitted.

Version 2:

Reviewer comments:

Reviewer #1

(Remarks to the Author)

The authors have now answered my main concern.

For the title, I suggest to remove "Through H3K4 trimethylation". The title would be:

The COMPASS subunit Bre2 targets CRF Arp9 to regulate the biosynthesis of secondary metabolite AFB1 and virulence in *Aspergillus flavus*.

Reviewer #3

(Remarks to the Author)

The revised manuscript has been substantially improved through the authors' careful consideration of the review comments and the addition of new experiments. In particular, the RT-qPCR analyses conducted on the SMAP (G4B84_002937) deletion strains have helped to clarify the pathway linking Bre2 to aflatoxin production.

However, several issues still require further attention.

Minor comments

1. Statistical analysis

In response to the question regarding statistical analysis (Reviewer #3, Major Comment 2), the authors answered:

“All the results in this study were expressed as the mean value of triplicates, and the error bars represented the standard deviation of at least three repetitions. The statistical analysis was performed using the software GraphPadPrism 10 (La Jolla, CA), and one-way ANOVA (one independent variable involved) or two-way ANOVA (more than one independent variable involved) was used to compare the statistical significance by Tukey's multiple comparisons test. In the manuscript, three groups (WT strain, deletion strain and complementary strain) were compared together, so Tukey's test is applicable.”

Two-way ANOVA is a method specifically designed for analyses involving two independent variables, not “more than one independent variable.” This description should therefore be corrected. In addition, ANOVA itself is not a method to “compare statistical significance”; rather, ANOVA assesses overall differences among groups, followed by post hoc tests such as Tukey's test to determine which specific groups differ.

Furthermore, with respect to the statistical analyses, the legend of Fig. 1 states:

“One-way ANOVA was used to compare the statistical significance by Tukey's multiple comparisons test for Panels B–D, F, J, and L. Two-way ANOVA was used to compare the statistical significance by Tukey's multiple comparisons test for Panels G and H.”

It is unclear how two-way ANOVA was applied to the results shown in Panels G and H. Did the authors treat time points (48 h and 72 h) as independent variables? If so, this approach may not be appropriate in this context. Instead, it would seem more suitable to perform one-way ANOVA separately at each time point (48 h and 72 h), using strain (WT, deletion strain, complementary strain) as the independent variable, followed by Tukey's post hoc test.

For all figures, the results of Tukey's test would be more appropriately presented using a compact letter display (alphabetical notation) rather than asterisks, since the use of asterisks can give the impression that multiple pairwise comparisons were performed repeatedly.

2. Figure 5B

In response to the question regarding Figure 5B (Reviewer #3, Minor Comment 6), the authors answered:

“To the peaks, we have not shown this clear in the original manuscript. In fact, the MS peaks in figure 5B do not indicate the number of proteins or peptides, but they reflect the separation resolution of the sample and the signal intensity of the peptides. The result of Figure 5B reflects that the tested samples are characterized with high separation resolution and signal intensity. And the protein samples in this study were digested by trypsin.”

The explanation referring to “separation resolution and signal intensity” does not appear to be sufficiently informative. Based on the presentation of Figure 5B—where the x-axis and y-axis appear to represent retention time and relative abundance, respectively, and the label “Base Peak FTMS + p NSI Full MS” is shown—the figure appears to represent a ion chromatogram obtained from LC–MS analysis using a full MS scan in positive ion mode over an m/z range of 300–1800.

Taken together, it seems that Figure 5B shows extracted ion chromatograms derived from LC/MS analysis of peptides obtained by trypsin digestion following immunoprecipitation of the target proteins. Is this interpretation correct? Please provide a precise and careful description of the experimental procedure in the figure legend.

3. Figure 7F

Regarding Figure 7F (Reviewer #3, Minor Comment 11), the authors answered:

“From the Fig.7F, according to the ‘percent of genes’, the genes in metabolism related KEGG categories (the purple color ones) accounts for 56%, so they are heavily involved in metabolism process. Related revision, according to your kind advice, is given in Line L501 (P22).”

It is unclear how the value of 56% was calculated. This number appears to represent the simple sum of genes classified under metabolism-related KEGG categories (assuming that the numbers shown on the bars represent gene counts). If so, expressing this value as a percentage may be misleading. Please clearly explain how the value of 56% was derived and describe the basis for this percentage calculation in the main text.

4. Selection of nine candidate genes

In response to the question regarding the criteria used to narrow down 214 downregulated genes to nine candidate genes (Reviewer #3, Minor Comment 13), the authors answered:

“214 downregulated genes were first filtered by the molecular-function of the GO enrichment, by which those related to fundamental molecular-level activities such as catalysis or binding, as well as those involved in the ordered biochemical or physiological processes at the cell, tissue or organism level listed under biological process were screened out ($P < 0.05$). Finally, the nine candidate genes were figured out in the second round of selection by KEGG pathway annotation for genes implicated in growth/development, reproduction, virulence, autophagy, proliferation, cell wall integrity, drug resistance, and immunity which are the key areas concerned by our lab.”

It is recommended that this explanation be incorporated into the main text (for example, around line 521) to ensure transparency and clarity regarding the gene selection process.

Version 3:

Reviewer comments:

Reviewer #3

(Remarks to the Author)

The authors' revisions have addressed most of the previous concerns, and the manuscript has been further improved. Only two minor points still require attention.

1. Statistical Analysis

As one-way ANOVA has been applied to panels G and H in Fig. 1, it appears that no analyses using two-way ANOVA remain in the manuscript. If this is the case, the description of two-way ANOVA should be removed from the Methods section.

2. Clarity of Expression

On page 25, line 530, the phrase “which are the key areas concerned by our lab” is unclear. This phrase should either be removed or replaced with a more objective expression.

Point-by-point response to the reviewer comments

Reviewer #1 (Remarks to the Author):

In this study, the authors first explore the consequences of inactivating the COMPASS subunit Bre2 in different biological processes of *A. flavus*. They show that Bre2 deletion ($\Delta bre2$) reduces the expression of key transcription factors and thereby affects fungal morphogenesis and the level of the secondary metabolite AFB1. In agreement with these results, they show that Bre2 is required for virulence *A. flavus* to silkworm. They further show that *A. flavus* is localized in the nucleus and that deletion or mutations in the SPRY phenocopy the $\Delta bre2$ deletion mutant. Mapping the peaks of H3K4me3 in WT reveals that this mark is mainly located in promoters and exons of genes notably involved in metabolite synthesis and in several genes of the SNF2 family. Interestingly, they show that Bre2 and Arp9 are each required for the expression of the one by the other. In the second part of the manuscript, the authors show that the deletion of Arp9 produces effects similar to the deletion of Bre2 indicating that Arp9 collaborates with Bre2 to modulate fungal morphogenesis and metabolite production. ATAC-seq experiments reveal that Arp9 acts by remodeling the promoter of target genes, including genes involved secondary metabolites related process.

1. In the paper, it is improper to write (lane 298):" H3K4me3 modification catalyzed by Bre2"; H3K4me3 is catalyzed by the Set1 complex

Answer: Thanks, we have revised the sentence according the professional advice.

It has been revised into "The above results indicated that H3K4me3 modification catalyzed by COMPASS with Bre2 as the key subunit plays an indispensable role in the fungal growth and development, response to stress, and secondary metabolites production."

The revision position was marked with purple background at **Line 318-319 (Page 13)**.

2. My major problem is with the title, the summary, and the role is assigned to Bre2. Actually, the observed effects are not specifically due to Bre2, but rather to the absence of H3K4me3 and the reduction of H3K4me2. It is likely that the deletion of *Sdc1* would produce the same effects as that of Bre2. The more appropriate title would be "H3K4me3 promotes CRF Arp9 expression to regulate the bio synthesis of secondary metabolite AFB1 and virulence in *Aspergillus flavus*".

Answer: Thanks for the thoughtful advice.

In fact, according to our pre-experiment, the phenotype of *sdc1* deletion strain is not the same as those of Bre2. It reflected that each subunit in the complex has its own characteristics. And in view of the *bre2* deletion strain was used as control strain in the ChIP-seq analysis, the title has been revised according to your professional advice to highlight the role of H3K4me3, and provided below:

"The COMPASS subunit Bre2 targets CRF Arp9 to regulate the bio-synthesis of secondary metabolite AFB1 and virulence by H3K4me3 in *Aspergillus flavus*"

It was marked with purple background at **Line 1 and 2 (Page 1)**.

3. In figure 1K, what are the neighboring peaks of the ABF1 peak which also disappear in the

Δ bre2 mutant? Specify the abbreviation TLC

Answer: The neighboring peaks of the ABF1 peak are AFG2, AFG1, and the AFB2 next to the AFB1. According to your kind hint, the abbreviation of TLC have been specified in **Line 142-143 (Page 5)** with purple background.

4. H3K4me3 ChIP-seq reveal 2871 enriched genes and 723 depleted genes (WT compared to the Δ bre2 strain). What are these 723 depleted genes?

Answer: Thanks, it is 723 down-regulated DAPs (differently accumulated peaks) in the WT samples compared to Δ bre2 samples. And we have check and revise it in **Line 307 (Page 13)** according to kind reminder with purple background.

5. Please provide the 63 proteins pulled down by Arp9.

Answer: Thank, we have provided the 63 proteins in an **Attached Excel File** titled with “63 proteins pulled down by Arp9”. As shown in **Line 387 (Page 16)**

Reviewer #2 (Remarks to the Author):

The manuscript describes regulation of a target to control the biosynthesis of secondary metabolites and virulence in *Aspergillus*. I appreciate the authors’ robust process used for genetic confirmations. The study is well presented and organized.

1. Grammatical clarity could be improved

Answer: Thanks, the grammars have been carefully improved according to your kind advice. Including Line 54, Line 66, Line 81, Line 92, Line 98, Line 99, Line 101, Line 112, Line 118, Line 131, Line 133, Line 153, Line 172, Line 173, Line 206, Line 207, Line 212, Line 217, Line 240, Line 245, Line 248, Line 251, Line 252, Line 259, Line 269-273, Line 294, Line 305, Line 306, Line 309, Line 313-315, Line 325, Line 330, Line 336, Line 342, Line 344, Line 345, Line 361, Line 364, Line 365, Line 372, Line 376, Line 380, Line 393, Line 397, Line 416-417, Line 430, Line 434-435, Line 488-489, Line 631, Line 640. They are all marked with purple background.

2. Define COMPASS and ATAC

Answer: Thanks, COMPASS is defined as “Complex of Proteins Associated with Set1” in **Line 70-71 (Page 3)**; and ATAC is defined as “**Assay for Transposase-Accessible Chromatin Using Sequencing**” in **Line 456 (Page 21)**.

3. Add reference to line 88 (i.e., Our group previously reported...)

Answer: Thanks, we have added related reference (**Refernce No. 35**) to the original line 88 which is **Line 96 (Page 4)** in the revision version with purple underground.

As shown in the Reference Section (**Line 1015-1016, Page 40**):

35. Liu, Y., Zhang, M., Xie, R., Zhang, F.&Zhuang, Z. J. F. i. M. The Methyltransferase AflSet1 Is Involved in Fungal Morphogenesis, AFB1 Biosynthesis, and Virulence of *Aspergillus flavus*. **11**,

(2020).

4. Line 125: italics to 'bre2'

Answer: Thanks, we have revised it in **Line 131 (Page 5)** (purple background) according to your kind advise.

5. Line 130: CM defined already?

Answer: Thanks, CM is defined as “6 g/L tryptone, 6 g/L yeast extract, 10 g/L glucose” in the **Line 738-739 (Page 32)** in the part of “Materials and methods”, and it is defined as “complete medium” in **Line 136 (Page 5)** according to your kind advise. And they are marked with purple background.

6. Line 136: TLC and HPLC defined? Check throughout if terms from methods are defined within the Results for first use

Answer: Thanks, TLC and HPLC were defined as “thin layer chromatography” and “high performance liquid chromatography” according to your kind advice in **Line 142-143 (Page 5)** with purple background.

And others are listed as below:

COMPASS (Complex of Proteins Associated with Set1) in **Line 70-71 (Page 3)**

CM (complete medium) in **Line 136 (Page 5)**

ATAC-seq (Assay for Transposase-Accessible Chromatin Using Sequencing) in **Line 456 (Page 21)**

7. Figure 1 – suggest increasing the size of the plots A-L (excluding K) to improve readability and decrease empty white space on the figures.

Most figures appear too small within the panels – sizes could be adjusted to optimize the figure size and reduce white space.

Answer: Thanks, we have adjusted the figures according to your kind advice. As shown in Figure 1 in Page 6; Figure 2 in Page 9; Figure 4 in Page 15; Figure 5 in Page 18; Figure 6 in Page 20; Figure 7 in Page 22; Figure 8 in Page 24.

Reviewer #3 (Remarks to the Author):

In the manuscript entitled "The COMPASS subunit Bre2 targets CRF Arp9 to regulate the bio synthesis of secondary 1 metabolite AFB1 and virulence in *Aspergillus flavus*", the authors focused on the gene of Bre2 subunit of the histone methyltransferase complex and showed that *bre2* is involved in H3K4 trimethylation in *A. flavus*. Through ChIP-seq analysis, they found that the gene of Arp9, which is involved in chromatin remodeling, is located downstream of *bre2*. The analysis of *bre2* and *arp9* knockout strains demonstrated that these genes are essential for the morphological development and aflatoxin production of *A. flavus*. Additionally, for *bre2*, they revealed functionally important protein domains and amino acid residues. Using epigenetic analysis of *arp9* mutants, they investigated downstream genes affected by Arp9-mediated chromatin remodeling, constructed knockout strains for some of these genes, and examined their phenotypes.

The authors utilized a variety of approaches including seq analysis on the generated mutant strains to elucidate the relationship between epigenetic modifications—initiated by histone H3K4 trimethylation—and aflatoxin production of *A. flavus*, the notorious mycotoxigenic fungus. The experimental results clearly establish that *arp9* functions downstream of *bre2* and suggest that Arp9 exerts a distal effect on the aflatoxin biosynthetic pathway and morphological development pathway through SMAP and other factors.

1. The manuscript does not fully elucidate the mechanism by which Bre2 and Arp9 regulates aflatoxin production and morphological development. Although the ATAC-seq analysis identified several genes as potential components of the Arp9 regulatory pathway, only SMAP knockout strains exhibited significant effects on aflatoxin production and morphology, leaving the relationship between SMAP and these factors unclear. Simply to say, in Figure S6, the arrow which leads to the red boxes enclosing the gene clusters is unidentified. Due to this limitation, the overall framework linking histone H3K4 trimethylation initiated by Bre2 to aflatoxin production has not been fully clarified. Further analyses are needed to bridge this missing link.

Answer: Thanks for your professional advice. To fully clarify the overall framework linking histone H3K4 trimethylation initiated by Bre2 to aflatoxin production and fungal morphological development, four more targets of Arp9 (including G4B84_002788, G4B84_009741, G4B84_009537 and G4B84_008657) were deeply analyzed in **Line 493-511 (Page 23)** with purple background, and **Fig. 8 (Page 24)**.

2. The authors concluded that Bre2 is related to the colonization of peanuts and maize based on experiments using the poorly growing deletion strain $\Delta bre2$. However, since poor growth is associated with colonization, it is not possible to discuss a direct relationship between Bre2 and colonization ability.

Thanks, to discuss a direct relationship between Bre2 and fungal colonization ability, the fungal virulence related factors, including *cp1*, *fbg1*, *lysM*, *mhp1*, *mpg1* and *tox1*, were monitored with qRT-PCR. The results showed that the absence of Bre2 significantly depressed the expression of these virulence factors, as shown in **Line 176-180 (Page 7)** and **Fig. S2A**.

3. Similarly, in the infection experiment with silkworms, the strong relationship between the poor growth of the deletion strain and its virulence makes it difficult to discuss the relationship between

Bre2 and virulence in silkworms.

Answer: Thanks, to deeper explore the virulence of Bre2 to animals, we constructed a *Galleria mellonella* larvae model. In the model, we compared the virulence of $\square bre2$ and WT stain to the immune system of *Galleria mellonella* larvae, and the results showed the absence of Bre2 serious affects the response of cellular-immunity and humoral-immunity of the larvae compared with the WT group, which reflected that Bre2 play an important role in fungal virulence to animal hosts. The revision part is in **Line 181-201 (Page 7-8)** with purple back ground and in **Figure 2G-2N (Page 9)**.

4. Additionally, it should be noticed that the relationship between fungal AF productivity and pathogenicity to silkworms is unclear. In fact, the $\Delta bre2$ strain still maintains lethality, with more than half of the silkworms dying within 6 days. As the same methods were used in experiments using other poorly growing deletion strains, no conclusions can be drawn from this paper regarding the relationship between Bre2 and virulence in *A. flavus*.

Answer: Thanks for your professional advice. To explore the role of AFs in the virulence of *A. flavus* to hosts, the silkworms were injected with equal volume of 10% DMSO dissolved AFs extracted from $\Delta bre2$, Com-*bre2* and WT strain each day. And as shown in **Fig. S2**, the results showed that the survival rate of $\Delta bre2$ group was significantly highly than that of Com-*bre2* and WT group, especially at 96 h and 120 h, which reflected that the concentration of AFs is one of the critical factors in fungal virulence against hosts regulated by Bre2. The revision is in **Line 195-201 (Page 8)** with purple background, and in **Fig. S2B and S2C**.

5. Figure 4A: Appears to be a screenshot from IGV, but lacks sufficient explanation.

Answer: Thanks, more detail explanation was added as “reflecting the trimethylation modification level of H3K4 on this chromatin fragment” to help readers understand the intended information according to your kind advice. And the revision is in the **Line 331-332 (Page 14)** with purple background.

6. Figure 5B: The meaning of the mass spectrum is unclear.

Answer: Thanks, more explanation- “the samples were sent for mass spectrometry identification in Applied Protein Technology Biotechnology Co., Ltd., and the interaction protein peaks of Arp9 were obtained as shown in **Fig.5B** and in the **Attached Excel File**, with WT sample as control.” in **Line 385-387 (Page 16)** was added to make the mass spectrum clear according to your kind advice.

7. Figure 5C-E: The differences between lanes are not explained, and it is difficult to read what the detected bands represent.

Answer: Thanks, the lanes have been clearly explained in **Line 399-402 (Page 17)** with purple background according to your kind advice.

It is “Western-blotting confirmed that in the Arp9-3HA strain, Strep was fused with RSC8 (**Fig. 5C**, two transformants of Arp9-3HA/RSC8-Strep strain), Sth1 (**Fig. 5D**, four transformants of Arp9-3HA/Sth1-Strep strain), Arp7 (**Fig. 5E**, five transformants of Arp9-3HA-Arp7-Strep strain) protein, respectively.”

8. Figure 5F-H: Additional explanations for the bands are needed.

Answer: Thanks, explanations for the bands have been added in **Line 402-414 (Page 17)** with purple background according to your kind advice.

It is shown as “Co-immunoprecipitation experiments showed that, in the Arp9-3HA/RSC8-Strep strain, Arp9-3HA could be pull-down, enriched and detected with anti-HA antibody, and anti-Strep antibody detection showed that RSC8 was interacted with Arp9 and pull-down together with Arp9 by anti-HA antibody (**Fig. 5F** above). The lower panel of **Fig. 5F** showed that, in the same fungal strain, RSC8-Strep could be pull-down, enriched and detected with anti-RSC8 antibody, while anti-HA antibody detection showed Arp9 was interacted with RSC8 and pull-down together with RSC8 by anti-Strep antibody (**Fig. 5F** below). Using the same method, it was found that Arp9 also interacts directly with Arp7 of the SWI/SNF complex, as shown in **Figure 5G**. Interestingly, after Arp9 was immunoprecipitated by anti-HA magnetic beads from Arp9-3HA/Sth1-Strep strain, in the final Co-IP analysis, Sth1-Strep could not be detected in the experimental group (**Fig. 5H**, the above two panels). However, when immunoprecipitation of Sth1 was carried out firstly with anti-strep magnetic beads, Arp9 could be detected with HA antibody in the experimental group (**Fig. 5H**, the lower two panels). ”

9. Figure 6A: The descriptions of the lanes and labels lack.

Answer: Thanks, the descriptions of the lanes and labels are provided in the legend of Figure 6A in **Line 446-449 (Page 20)** with purple background as below: “Compared to WT strain, the *bre2* ORF and *arp9* ORF could not be amplified from the $\Delta bre2\Delta arp9$ strain, while AP and BP fragments could amplified from $\Delta bre2\Delta arp9$ strain, but could not detected from WT. The DNA fragments amplified from WT is bigger than that form the $\Delta bre2\Delta arp9$ strain with nested primers Bre2-pyrG-NF and Bre2-pyrG-NR.”, which confirmed that the $\Delta bre2\Delta arp9$ strain had been successfully constructed.

10. Figure 7: The text is too small to read. The labeling may be inappropriate (e.g., "arp9" should be " $\Delta arp9$ ", isn't it?). The same issue applies to Figure S3.

Answer: Thanks, the text of Figure 7 has been enlarged according to your nice advice.

Yes, *arp9* in this figure should be " $\Delta arp9$ ", and we have added the corresponding explanation/descriptions in the end of the legend in **Line 485 (Page 22)** with purple background.

11. Figure 8A: The method for calculating the CAI index is unclear. Is there an easier metric to understand?

Answer: Thanks, we have introduced the method for calculating the CAI index according to the original reference (Gonzalez, R, and C Scazzocchio. “A rapid method for chromatin structure analysis in the filamentous fungus *Aspergillus nidulans*.” *Nucleic acids research* vol. 25,19 (1997): 3955-6. doi:10.1093/nar/25.19.3955) in the Methods section in **Line 864-869 (Page 36)** with purple background according to your professional advice.

It is shown as “Due to this characteristic, the region where DNaseI is prone to digestion is the area where chromatin is open. qPCR analysis of the DNaseI-treated samples was performed to measure the relative abundance (accessibility) of target regions. The results were calculated by $2^{-\Delta\Delta C_t}$ and normalized with the housekeeping gene - tubulin, then the final result is the relative chromatin accessibility index (CAI). Primer sequences are listed in **Table S4**.”

12. Line 40-42: As relationship of AF productivity of *A. flavus* with its pathogenicity in humans has not been proved, this sentence should be separated to avoid misunderstandings.

Answer: We agree, and we have revised the sentence in **Line 46-48 (Page 2)** with purple background according to your professional advice.

It is revised into “Due to producing the most toxic secondary metabolite-aflatoxins (AFs), the notorious *Aspergillus flavus* poses a serious threat to human health by food contamination”

13. Figure 1I: Spots are overlapped. It necessary to show the AFB1 spot clearly.

Answer: Thanks, we have repeated the TLC in **Figure 1I (Page 6)** according to your nice advice.

14. Figure 2H: It’s necessary to show the data of a week.

Answer: Thanks, to deeper explore the virulence of Bre2 to animals, we constructed a *Galleria mellonella* larvae model in **Line 181-195 (Page 7-8)** with purple background and the **Figure 2G-2N in Page 9**. And the data of the experiment was collected for a week following your kind advice.

15. Figure 2K: Rf values are different among the spots.

Answer: Thanks, to deeper explore the virulence of Bre2 to animals, we constructed a *Galleria mellonella* larvae model. And the quality of Figure 2K, **Figure 2N (Page 9)** in the revision version was improved according to your professional advice.

16. Line 215: The website should be included in the Methods section.

Answer: Thanks, the website "MEME (<https://meme-suite.org/meme/tools/meme>)” has been included in the Methods section in **Line 751 (Page 32)** according to your nice advice.

17. Figure 3A: The intensity of the H3 bands varies greatly between lanes, making it unsuitable as a control.

Answer: Thanks, the WB has been repeated according to your kind advice, and the intensity of the H3 bands is stable in the new WB analysis in the **Figure 3A (Page 12)**.

18. Line 313-314: The statement "the enrichment level in WT was significantly higher (more than twice times)" lacks numerical data, making it unclear.

Answer: Thanks, we have changed it into “the enrichment level of this chromatin fragment from WT is $2^{1.1}$ times ($P < 0.05$) of that from $\Delta bre2$ strain (Fig. 4A)” in **Line 333-334 (Page 14)** with purple background.

19. Line 363: The sentence "Immunoprecipitation was used to enrich ~" is unclear because Figure 5A presents a Western blot result, and it is not evident what was enriched.

Answer: Thanks, we have made our description more precisely in **Line 382-384 (Page 16)** according to your kind advice.

It is revised into “Immunoprecipitation was used to enrich the proteins interacting with Arp9, and the outcome of the enrichment (more HA tagged Arp9 was enriched in Arp9-3HA strain of IP group than the fungal strain in the Input group) was confirmed by western blotting as showed in

Fig. 5A”.

20. Line 364: Information about the outsourced company should be placed in the Methods section.

Answer: Thanks, the detail information about the outsourced company “Applied Protein Technology Biotechnology Co., Ltd. (Shanghai, China)” has been placed in the Methods section in **Line 878-879 (Page 37)** with purple background.

21. Lines 365-367: There is an inconsistency between Line 365, which states "totally 63 proteins," and Line 367, which states "64 proteins." Additionally, it would be helpful to provide a table listing these proteins including the five screened-out proteins.

Answer: Thanks, it should be 63 proteins, and after add Arp9 itself, it become 64, we have revised it in **Line 387 and 389 (Page 16)** in purple background.

And we have provided the list of these proteins in an **Excel file titled “63 proteins pulled down by Arp9”** as one of the attachment materials according to your professional advice.

22. Line 370: What is the reasoning for stating that “~was found to tightly interact with Arp9”?

Answer : Thanks, we have revised the sentence into “RSC8 is a subunit in the SWI/SNF complex, however, it has not been reported whether there is a direct interaction between Arp9 and RSC8” in **Line 390-392 (Page 16)** in purple background.

23. Lines 374-375: It is unclear how the dual-tagged *A. flavus* was constructed and verified. In addition, the company name should be included in the Methods section.

Answer: Thanks, in this experiment, we constructed dual-tagged strains by homologous recombination. Firstly, the 3HA-*pyrG* gene fragment was ligated behind the *Arp9* gene through fusion PCR. Then, to obtain Apr9-3HA single-tag strain by transforming protoplasts of *A. flavus*-CA14 PTS with amplified Apr9-3HA-*pyrG* fragment. After the *pyrG* gene was deleted from Apr9-3HA single-tagged strain under the stress of 5-FOA, RSC8-Strep-*pyrG* fragment was introduced into the protoplasts of the Apr9-3HA single-tagged strain by homologous recombination, and the Apr9-3HA/RSC8-Strep dual-tagged strain was screened with the resuscitation medium without uracil and uridine. Apr9-3HA/Sth1-Strep dual-tagged strain and Arp9-3HA/Arp7-Strep dual-tagged strain were prepared with the same method. Then, the dual-tagged *A. flavus* strains were verified by sequencing in Beijing Tsingke Biotech Co., Ltd., and western-blotting. And the company name has been included in the Methods section in **Line 783** according to your kind advice. All related information was provided in **Line 774-784 (Page 33)** in purple background.

24. Line 408: Does this statement align with the observation that *bre2* gene expression was reduced in the *arp9* knockout strain?

Answer: Thanks, we have made corresponding revision in **Line 441-442 (Page 19)** according to your nice reminder.

It has been revised into “All the above results suggested that Bre2 and Arp9 are deeply involved in hyphal growth and conidiation, indispensable in sclerotia formation and AFB1 bio-synthesis, and Arp9 might be downstream of Bre2 in the Bre2 triggered epigenetic signaling

pathway, additionally, they might establish a positive-feedback in view of that Arp9 dramatically promotes the expression of *bre2* (Fig 4C).”

25. Lines 451-452: Explanation is need for the criteria for selecting target genes in ATAC-seq analysis. Were they screened based on GO terms?

Answer: Yes, the criteria for selecting target genes in ATAC-seq analysis is the possibility of being a virulence factor in *A. flavus* according to GO terms, and we have added the explanation in **Line 488-489 (Page 23)**.

It is revised into “To further reveal the signaling pathway mediated by Arp9, the targets potentially regulating fungal development, secondary metabolites bio-synthesis and pathogenicity according to GO terms are screened out from the results of ATAC-seq analysis”

26. Line 456: As mentioned above, the chromatin accessibility index is not well defined. The Methods section only mentions that it was calculated using the $2^{-\Delta\Delta CT}$ method, which is typically used for qPCR quantification.

Answer: Thanks, we have defined the chromatin accessibility index according to the original reference (Gonzalez, R, and C Scazzocchio. “A rapid method for chromatin structure analysis in the filamentous fungus *Aspergillus nidulans*.” *Nucleic acids research* vol. 25,19 (1997): 3955-6. doi:10.1093/nar/25.19.3955) in the Methods section in **Line 864-869 (Page 36)** according to your professional advice.

It is changed into “Due to this characteristic, the region where DNaseI is prone to digestion is the area where chromatin is open. qPCR analysis of the DNaseI-treated samples was performed to measure the relative abundance (accessibility) of target regions. The results were calculated by $2^{-\Delta\Delta Ct}$ and normalized with the housekeeping gene - tubulin, then the final result is the relative chromatin accessibility index (CAI). Primer sequences are listed in **Table S4**.”

Reviewer #4 (Remarks to the Author):

Answer: Thanks, we have carefully and cautiously revised our manuscript according to the professional advice provided by all reviewers.

Point-by-point response to the reviewer comments

REVIEWER COMMENTS

Reviewer #1 (Remarks to the Author):

The authors did not respond to my major criticism regarding the specific role that Bre2 could play in relation to another subunit of the Se1 complex, such as Sdc1. To validate a number of the article's conclusions, it is essential to compare and show in the article the differences in phenotypes between *bre2*Δ and *sdc1*Δ.

Answer: Thanks for your professional advice. The *Δsdc1* strain is constructed and its phenotype is compared with that of *Δbre2* according to your kind advice. The results showed that, different from Bre2, Sdc1 participates in fungal morphogenesis, but not involved in the regulation of AFB1 biosynthesis, and the related contents are shown in **L633-639 (P 30)** with blue background and in **Fig. S9**.

The new title is incomprehensible.

Answer: Thanks, the title has been revised according to your kind advice. It is as below now: **“The COMPASS subunit Bre2 targets CRF Arp9 to regulate the biosynthesis of secondary metabolite AFB1 and virulence through H3K4 trimethylation in *Aspergillus flavus*”**

Reviewer #2 (Remarks to the Author):

The authors have addressed my points; I appreciate the grammatical clarity and figure revisions.

Answer: We sincerely appreciate your professional advice and valuable reminders, which will greatly facilitate the quality improvement of our manuscript.

Reviewer #3 (Remarks to the Author):

The manuscript has been substantially improved by the addition of new experiments, revisions of the text, and significant improvements to the figures (including enlargement of labels). These efforts have strengthened the work overall. However, several major issues remain that should be addressed.

Answer: Thanks very much for your professional reminders and suggestions for the improvement of the quality of our manuscript.

Major comments:

1. Four additional genes were proposed as putative targets of Arp9, and corresponding knockout mutants were generated and analyzed. However, aside from Stromal membrane-associated protein (SMAP, G4B84_002937), none of the knockouts appear to have a substantial impact on aflatoxin

production.

The role of SMAP in aflatoxin biosynthesis, fungal growth, and conidiation has not been previously reported and remains unclear. The mechanism underlying the observed effect of SMAP deletion is entirely unknown. In the Discussion, the authors only mention its involvement in stroma-supported erythropoiesis and erythropoietic activity in mice, which has no apparent relevance to *Aspergillus flavus*. The authors should therefore provide at minimum a plausible mechanistic hypothesis, and ideally, supporting validation experiments.

Answer: Thanks for your professional advice. We agree and have performed further RT-qPCR to explore the regulatory mechanism of SMAP, with relevant details presented in **L763-765 (P35)** of Discussion section with blue background and in **Fig. S8**.

2. In the Materials and Methods, the Data Analysis section initially states that “All data were expressed as mean \pm standard deviation,” but later specifies that “Error bars represented the standard error of at least three repetitions.” Which is correct?

Furthermore, the section indicates that statistical analysis was performed using Tukey’s test. However, Tukey’s test is not appropriate in all cases (e.g., for comparisons between two groups). It is also unclear whether proper attention was paid to the issue of multiple comparisons. Statistical methods should be described not only in Materials and Methods but also clarified in each relevant figure legend where statistical data are presented.

Answer: Thanks for this professional reminder. We have revised related contents and figures according to your advice. As the revision in the Section of “**Materials and methods**” in **L919-926 (P40)**: All the results in this study were expressed as the mean value of triplicates, and the error bars represented the standard deviation of at least three repetitions. The statistical analysis was performed using the software GraphPadPrism 10 (La Jolla, CA), and one-way ANOVA (one independent variable involved) or two-way ANOVA (more than one independent variable involved) was used to compare the statistical significance by Tukey’s multiple comparisons test. In the manuscript, three groups (WT strain, deletion strain and complementary strain) were compared together, so Tukey’s test is applicable. The unpaired two-tailed t-test was used to compare the statistical significance between two groups. The log-rank test was used in the comparison of the difference in survival rates between animal injection groups in median survival time. And statistical significance was determined as * ($P < 0.05$), ** ($P < 0.01$) and *** ($P < 0.001$).

Statistical methods were briefly introduced in each relevant figure legend in **L166-168 (P7)**, **L246-248 (P11)**, **L302-303 (P13)**, **L370-371 (P16)**, **L480-481 (P22)**, **L565-567 (P27)**, and the legends of **Fig. S1**, **Fig. S2**, **Fig. S3**, **Fig. S6**, **Fig. S7**, **Fig. S8**, and **Fig. S9** with blue background.

3. It is not appropriate to directly compare the virulence of the WT strain and the Δ bre2 strain when there is a significant difference in their growth. The authors argue that the Δ bre2 strain exhibits reduced virulence because it has a weaker impact on the immune system of the larvae compared to the WT strain. However, this reduced impact may simply be due to the impaired growth of the Δ bre2 strain, rather than a direct consequence of the Bre2 deletion. Additionally, the authors suggest that the survival rate of larvae injected with extracts from the WT and Δ bre2 strains is related to the amount of AF contained in those extracts. To support this claim, it is necessary to conduct experiments using authentic AF at varying concentrations, and to quantify the AF content in the extracts before repeating the injection experiments.

Answer: Thanks for this professional reminder, and we have performed fungal spore injection experiment to explore if the reduced virulence of $\Delta bre2$ strain just due to the impaired growth of the $\Delta bre2$ strain in **Fig. S3**, and the results demonstrated that fungal mycelium loading (reflecting the growth of fungal hyphae) is not the key factor in the reduced virulence of $\Delta bre2$ strain as showed in **L205-213 (P8)**.

The pre-experiments for AF content quantification had been performed prior to the presented AF-injected larvae results, and these pre-experiment data for AF quantification are presented in **Fig. S2B and S2C** and in **L194-202 (P8)** according to your professional advice.

4. There remain numerous grammatical errors, especially in the newly added texts with purple highlight (e.g., incorrect verb tenses). The manuscript should undergo professional English editing.

Answer: Thanks, the grammatical errors have been carefully corrected according to your kind reminder, and they are marked with blue background in **Line 31, 35, 53, 80-82, 91, 101, 132, 139, 164, 189, 204, 223, 226, 227, 237, 241, 244, 256, 257, 265, 274, 283, 288, 317, 323, 341, 365, 368, 376, 387, 394, 395, 449, 457, 465, 471, 525, 557, 571-573, 580, 593, 597, 600, 611, 617, 625, 678, 727, 729, 761, 763, 766, 871, 893, and 906**.

Minor comments:

1. Title: The expression “by H3K4me3” is unclear. In addition, abbreviations such as “me3” should be avoided in the title.

Answer: Thanks for your advice, “me3” has been fully expressed as “trimethylation”, and the title has been revised into “The COMPASS subunit Bre2 targets CRF Arp9 to regulate the biosynthesis of secondary metabolite AFB1 and virulence **through H3K4 trimethylation** in *Aspergillus flavus*”

2. It would be appropriate to mention SMAP in the summary.

Answer: Thanks for your professional advice, SMAP has been appropriately mentioned in the summary in **Line 38 (P2)** with blue background, as “ATAC results revealed that mainly through SMAP, Arp9 deeply participates in fungal pathogenicity through regulating a serial secondary metabolism, morphogenesis and virulence associated regulators by modulating the chromatin conformation.”

3. Line 47: The primary cause of aspergillosis is *A. fumigatus* rather than *A. flavus*. The relevance of the present study to aspergillosis is uncertain, and therefore this description should be deleted.

Answer: Thanks, we have deleted this description, and the revised sentence is marked with blue background in **Line 46-47 (P2)**.

4. Line 334: In “the enrichment level of this chromatin fragment from WT is $2^{1.1}$ times ($P < 0.05$) of that from $\Delta bre2$ strain,” it is unclear how the enrichment level was calculated, and thus the meaning of “2.1.1 times” is ambiguous. Moreover, without information on sample size and the specific statistical test applied, the interpretation of reported P value is uncertain.

Answer: Thanks for the professional reminder. It has been checked and corrected and revised into “the enrichment level of this chromatin fragment from WT is $2^{1.1}$ times ($FDR < 0.05$) of that from

Δbre2 strain.” And the enrichment levels were calculated with peak areas. It is showing in **L347-349 (P15)**.

5. Line 350 (Figure 4): The label “H3K36me3” appears to be a mistake; it should likely read “H3K4me3.”

Answer: Thanks, we have corrected it into “H3K4me3” in **L365 (P16)** according to your nice reminder.

6. Lines 385–386 and Figure 5B: The name of the company should be reported in the Methods section, while the description here should focus on clarifying what experiments were performed and how the data were analyzed. In particular, Fig. 5B raises questions: based on the MS peaks of Arp9-3HA and WT, it seems unlikely that Arp9-3HA contains peaks corresponding to 63 additional proteins compared with WT. Were these peaks derived from peptides following tryptic digestion, or from another protease?

Answer: Thanks for the kind advice. The name of the company has been given in **L916-917 (P40)** in the Methods section. And related experiments are further clarified in **L398-402 (P17)** according to your professional advice.

To the peaks, we have not shown this clear in the original manuscript. In fact, the MS peaks in figure 5B do not indicate the number of proteins or peptides, but they reflect the separation resolution of the sample and the signal intensity of the peptides. The result of Figure 5B reflects that the tested samples are characterized with high separation resolution and signal intensity. And the protein samples in this study were digested by trypsin. We have revised and improved these in this version (**L399-401, P17**).

7. Line 390: The criteria by which the five proteins were “screened out” should be clearly defined.

Answer: Through joint analysis with the STRING database (a tool for predicting functional protein association networks; string-db.org), the Arp9 interacting protein, RSC8, was screened out from these 63 candidates based on the criterion that the enrichment levels in the Arp9-HA strain were fourfold higher than those in the WT strain. And in the chromatin remodeling complex SWISNF, Arp9 is highly conserved and needs to form a heterodimer with another subunit of the complex, Arp7, to function. Hence, it is speculated that there is an interaction between Arp7 and Arp9. Similarly, the core subunit of the chromatin remodeling complex SWISNF, Sth1, is central to the function of the complex, and a heterodimer of Arp9 and Arp7 can bind to the HAS domain of Sth1. Thus, we speculate that there is an interaction between Arp9 and core subunit Sth1, which plays a chromatin remodeling role. Therefore, Co-IP analysis was carried out to further verify whether arp9 interacted with RSC8, Arp7 and Sth1. The related introduction is presented in **L402-411 (P17)**.

8. Lines 397–414: This section contains many grammatical errors and requires thorough editing. A concise explanation of the “Strep” tag is also needed.

Answer: Thanks, the grammatical errors from **L394 to L432 (P17-18)** have been thoroughly revised according to your professional advice. And the “Strep” is a tag composed of 8 amino acids

that can specifically bind to Streptavidin or its derivative StrepTactin. It is widely used in protein purification, immunoprecipitation, and protein-protein interaction research. The “Strep” tag is concisely explained in **L412-413 (P17-18)** as: “Arp7 further labeled with eight-amino-acid Strep tag that can specifically bind to Streptavidin or its derivative StrepTactin”.

9. Figure 5 (F–H): Each panel requires a more detailed explanation. What detection methods were used—silver staining or Western blotting (and if Western blotting, with which antibody)? What proteins do the detected bands represent? In panel H, the band sizes are not indicated.

Answer: Thanks for your kind advice. The detection methods were Western blotting, and each panel was given a detailed explanation. The band sizes in Panel H have been indicated, and it has been revised as follows in **L438-447 (P19)** according to your professional advice:

(F) The upper panel western-blotting results were obtained from the Arp9-3HA/RSC8-Strep strain: after IP with anti-HA magnetic beads, Arp9 (100 kDa) was detected using an anti-HA antibody, and RSC8 (100 kDa) using an anti-Strep antibody; the lower panel shows IP with anti-Strep beads, followed by western-blotting detection of RSC8 using an anti-Strep antibody and Arp9 using an anti-HA antibody. (G) The upper panel shows western-blotting results from the Arp9-3HA/Arp7-Strep strain: after IP with anti-HA magnetic beads, Arp9 was detected using an anti-HA antibody, and Arp7 (70 kDa) using an anti-Strep antibody. The lower panel was IP with anti-Strep beads, followed by western-blotting detection of Arp7 via an anti-Strep antibody, and Arp9 via an anti-HA antibody. (H) The upper panel shows western-blotting results from the Arp9-3HA/Sth1-Strep strain: After IP with anti-HA magnetic beads, Arp9 was detected using an anti-HA antibody and Sth1 (180 kDa) using an anti-Strep antibody. The lower panel presents IP with anti-Strep beads, followed by western-blotting detection of Sth1 via an anti-Strep antibody, and Arp9 via an anti-HA antibody. The band sizes in panel H are indicated according to your kind reminder.

10. Figure 6A: The meaning of “AP” and “BP” should be explained.

Answer: Thanks, we have revised it according to your kind advice, as following in **L473-475 (P21-22)**: The "AP" fragment, which overlaps the 5' flanking sequence and *pyrG*, was amplified with the validation primers Arp9-AP-F and Arp9-AP-R. The "BP" fragment, overlapping *pyrG* and the 3' flanking sequence, was amplified by primers Arp9-BP-F and Arp9-BP-R.

11. Lines 473–476: From Fig. 7F, it does not appear that “metabolism, oxidative phosphorylation, and fatty acid degradation” are particularly heavily enriched. Rather, a variety of KEGG categories seem to be represented. What is the basis for concluding that these specific categories are heavily enriched?

Answer: Thanks to this professional advice. From the **Fig.7F**, according to the “percent of genes”, the genes in metabolism related KEGG categories (the purple color ones) accounts for 56%, so they are heavily involved in metabolism process. Related revision, according to your kind advice, is given in Line **L501 (P22)**.

12. Line 478: The statement “especially oxidative and secondary metabolites related process” does not seem well supported by the information in Fig. 7. A more detailed explanation of results of the

KEGG and GO term enrichment is required.

Answer: Thanks for the professional advice. The related content is revised as “Further KEGG pathways analysis showed that these down-regulated DAPs genes of $\Delta Arp9$ were heavily involved (accounts for 56% genes in KEGG, and 73% GO annotated genes) in the process of metabolism (including Lipid metabolism, Carbohydrate metabolism, Amino acid metabolism, Nucleotide metabolism, Metabolism of terpenoids and polyketides, and Biosynthesis of other secondary metabolites.)”. It is shown in **L500-504 (P22-23)** according to your kind advice.

13. Line 490: The basis for narrowing down 214 downregulated genes from the ATAC-seq results to nine candidate genes should be described. From which GO terms were they selected? Were there other genes belonging to the same GO terms that were excluded?

Answer: 214 downregulated genes were first filtered by the molecular-function of the GO enrichment, by which those related to fundamental molecular-level activities such as catalysis or binding, as well as those involved in the ordered biochemical or physiological processes at the cell, tissue or organism level listed under biological process were screened out ($P < 0.05$). Finally, the nine candidate genes were figured out in the second round of selection by KEGG pathway annotation for genes implicated in growth/development, reproduction, virulence, autophagy, proliferation, cell wall integrity, drug resistance, and immunity which are the key areas concerned by our lab.

14. Line 511: “Fig. 8M” likely refers to “Fig. 8O.” Additionally, the phrase “significantly decreased” should be reserved for statistically significant differences. Although the AFB1 spot appears weaker, the criteria for significance are not provided.

Answer: Thanks for your kind reminder, it has been corrected into “**Fig. 8O**” in **L539 (P25)**. And significant analysis among the AFB1 spots has been provided in **Fig. 8P (P26)** according to your professional advice.

15. Figures 8J and 8L: Each panel shows two plate images (top and bottom), but this is not described in the figure legends. Furthermore, in panel L, there should be no “empty position,” yet the legend mentions.

Answer: The top, middle and bottom plates are described in the figure legends according to your kind reminder as following from **L555-557** and **L559-560 (P27)**: (J) The spores of $\Delta G4B84_010461$, $\Delta G4B84_002937$ and $\Delta G4B84_005528$ were incubated onto CM media under 37°C for 7 d (above panels); Middle panels were sprayed with 75% ethanol, and the lower panels were the enlarged pictures of the middle panels under a dissecting microscope. (L) The spores of $\Delta G4B84_002788$, $\Delta G4B84_009741$, $\Delta G4B84_009537$ and $\Delta G4B84_008657$ were incubated onto CM media under 37°C for 7 d (above panels); Middle panels were sprayed with 75% ethanol, and the lower panels were the enlarged pictures of the middle panels.

And thanks, we have removed the unnecessary content from the legend of panel L according to your kind remainder as shown in **L559 (P27)**.

16. Discussion: I suggest that the Discussion be made more concise, focusing on the insights derived from the findings rather than merely summarizing the results.

Answer: Thanks for the professional advice, we have made the Discussion more concise and

focusing on the insights in **L595-596 (P29)**, **L633-639 (P30)**, **L644-646 (P30-31)**, **L658-659 (P31)**, **L663-665 (P31)**, **L686-688 (P32)**, **L714-716 (P33)**, **L763-765 (P35)**.

17. Figure S7: This figure is important for understanding the manuscript, and I recommend that it be moved from the Supplementary section to the main figures (e.g., Figure 9). However, several points require clarification. Currently, it appears that all nine genes affected by Arp9, including SMAP, influence the aflatoxin gene cluster and other genes (red boxes). In reality, their effects are likely to differ. For example, gene G4B84_008657 (DAK) clearly does not affect aflatoxin production. It would be preferable to draw arrows only from the strongly influential genes, such as SMAP. In addition, the term “other biosynthetic gene clusters” is vague and may be better omitted.

Answer: We agree with your opinion. Figure S7 has been moved from the Supplementary section to the main figures, as the **Fig. 9** in **Page 28**. Arrow is drawn from the strongly influential gene, SMAP. The term “other biosynthetic gene clusters” has been deleted according to your kind advice.

** See Nature Portfolio’s author and referees' website at www.nature.com/authors for information about policies, services and author benefits.

Point-by-point response to the reviewer comments

REVIEWER COMMENTS

Reviewer #1 (Remarks to the Author):

The authors have now answered my main concern.

For the title, I suggest to remove "Through H3K4 trimethylation". The title would be:

The COMPASS subunit Bre2 targets CRF Arp9 to regulate the biosynthesis of secondary metabolite AFB1 and virulence in *Aspergillus flavus*.

Answer: We agree, and we have revised the title (The COMPASS subunit Bre2 targets CRF Arp9 to regulate the biosynthesis of secondary metabolite AFB1 and virulence in *Aspergillus flavus*) accordingly to your kind advice with purple background (See Line 1-2, Page 1).

Reviewer #3 (Remarks to the Author):

The revised manuscript has been substantially improved through the authors' careful consideration of the review comments and the addition of new experiments. In particular, the RT-qPCR analyses conducted on the SMAP (G4B84_002937) deletion strains have helped to clarify the pathway linking Bre2 to aflatoxin production.

However, several issues still require further attention.

Minor comments

1. Statistical analysis

In response to the question regarding statistical analysis (Reviewer #3, Major Comment 2), the authors answered:

"All the results in this study were expressed as the mean value of triplicates, and the error bars represented the standard deviation of at least three repetitions. The statistical analysis was performed using the software GraphPadPrism 10 (La Jolla, CA), and one-way ANOVA (one independent variable involved) or two-way ANOVA (more than one independent variable involved) was used to compare the statistical significance by Tukey's multiple comparisons test. In the manuscript, three groups (WT strain, deletion strain and complementary strain) were compared together, so Tukey's test is applicable."

Two-way ANOVA is a method specifically designed for analyses involving two independent variables, not "more than one independent variable." This description should therefore be corrected. In addition, ANOVA itself is not a method to "compare statistical significance"; rather, ANOVA assesses overall differences among groups, followed by post hoc tests such as Tukey's test to determine which specific groups differ.

Answer: Thanks for the professional reminder, we have revised “more than one independent variable” into “two independent variables” in Line 934 Page 41. And the relationship between ANOVA and Tukey’s test was made clearer in Line 934-937 Page 41 according to your nice advice, as below shown in purple background:

“The statistical analysis was performed using the software GraphPadPrism 10 (La Jolla, CA), and one-way ANOVA for one independent variable involved or two-way ANOVA for two independent variables involved was used to assess overall differences among groups (Determining whether significant differences exist among all intergroup differences), and the statistical significance between specific groups was further compared by Tukey’s multiple comparisons test.”

Furthermore, with respect to the statistical analyses, the legend of Fig. 1 states: “One-way ANOVA was used to compare the statistical significance by Tukey’s multiple comparisons test for Panels B–D, F, J, and L. Two-way ANOVA was used to compare the statistical significance by Tukey’s multiple comparisons test for Panels G and H.”

It is unclear how two-way ANOVA was applied to the results shown in Panels G and H. Did the authors treat time points (48 h and 72 h) as independent variables? If so, this approach may not be appropriate in this context. Instead, it would seem more suitable to perform one-way ANOVA separately at each time point (48 h and 72 h), using strain (WT, deletion strain, complementary strain) as the independent variable, followed by Tukey’s post hoc test.

Answer: Thanks, the treat-time-points (48 h and 72 h) are not the independent variables, they were analyzed with one-way ANOVA separately at each time point (48 h and 72 h). The related content has been revised as “One-way ANOVA was used to assess the overall statistical significances among groups, and Tukey’s multiple comparisons test was further performed for post-hoc analysis for Panel B-D, F-H, J and L.” (See Line 166-168, Page 7) in purple ground according to your professional reminder.

For all figures, the results of Tukey’s test would be more appropriately presented using a compact letter display (alphabetical notation) rather than asterisks, since the use of asterisks can give the impression that multiple pairwise comparisons were performed repeatedly.

Answer: Thanks, the asterisks in the results of Tukey’s test in all figures have been replaced with lowercase letters according to your kind advice. (See Figure 1, Figure 2, Figure 3, Figure 4, Figure 6, Figure 8, Figure S1, Figure S2, Figure S3, Figure S6, Figure S7, Figure S8, Figure S9).

2. Figure 5B

In response to the question regarding Figure 5B (Reviewer #3, Minor Comment 6), the authors answered:

“To the peaks, we have not shown this clear in the original manuscript. In fact, the MS peaks in figure 5B do not indicate the number of proteins or peptides, but they reflect the separation resolution of the sample and the signal intensity of the peptides. The result of Figure 5B reflects that the tested samples are characterized with high separation resolution and signal intensity. And the protein samples in this study were digested by trypsin.”

The explanation referring to “separation resolution and signal intensity” does not appear to be sufficiently informative. Based on the presentation of Figure 5B—where the x-axis and y-axis appear to represent retention time and relative abundance, respectively, and the label “Base Peak FTMS + p NSI Full MS” is shown—the figure appears to represent a ion chromatogram obtained from LC–MS analysis using a full MS scan in positive ion mode over an m/z range of 300–1800.

Taken together, it seems that Figure 5B shows extracted ion chromatograms derived from LC/MS analysis of peptides obtained by trypsin digestion following immunoprecipitation of the target proteins. Is this interpretation correct? Please provide a precise and careful description of the experimental procedure in the figure legend.

Answer: Thanks for this professional advice. We have revised it into “The target proteins obtained from immunoprecipitation were digested with trypsin, and the peptide segments were analyzed using LC-MS/MS (nanoLC QE) after enzymatic hydrolysis. The data was collected in positive ion mode, and the ion chromatogram (the peak chromatogram) was obtained by full MS scanning (m/z range 300-1800).” (See Line 439-441, Page 19) according to your kind advice.

3. Figure 7F

Regarding Figure 7F (Reviewer #3, Minor Comment 11), the authors answered:

“From the Fig.7F, according to the ‘percent of genes’, the genes in metabolism related KEGG categories (the purple color ones) accounts for 56%, so they are heavily involved in metabolism process. Related revision, according to your kind advice, is given in Line L501 (P22).”

It is unclear how the value of 56% was calculated. This number appears to represent the simple sum of genes classified under metabolism-related KEGG categories (assuming that the numbers shown on the bars represent gene counts). If so, expressing this value as a percentage may be misleading. Please clearly explain how the value of 56% was derived and describe the basis for this percentage calculation in the main text.

Answer: Thanks, we recalculated the percentage according to your nice advice. The number of genes in the metabolism related KEGG pathway divided by the total number of KEGG annotated genes is 60%, so the number of genes in the metabolic related KEGG pathway accounts for 60%

of the total KEGG annotated genes. It is described as “(accounts for 60% genes in KEGG (the number of genes in the metabolism related KEGG pathway divided by the total number of KEGG annotated genes)” (See Line 506-508, Page 22).

4. Selection of nine candidate genes

In response to the question regarding the criteria used to narrow down 214 downregulated genes to nine candidate genes (Reviewer #3, Minor Comment 13), the authors answered:

“214 downregulated genes were first filtered by the molecular-function of the GO enrichment, by which those related to fundamental molecular-level activities such as catalysis or binding, as well as those involved in the ordered biochemical or physiological processes at the cell, tissue or organism level listed under biological process were screened out ($P < 0.05$). Finally, the nine candidate genes were figured out in the second round of selection by KEGG pathway annotation for genes implicated in growth/development, reproduction, virulence, autophagy, proliferation, cell wall integrity, drug resistance, and immunity which are the key areas concerned by our lab.”

It is recommended that this explanation be incorporated into the main text (for example, around line 521) to ensure transparency and clarity regarding the gene selection process.

Answer: Thanks, we have incorporated the related content into the main text in Line 523-530 Page 25 according to your kind advice as below in purple background:

“To further reveal the signaling pathway mediated by Arp9, 214 downregulated genes were first filtered according to the molecular-function of the GO enrichment, by which those related to fundamental molecular-level activities such as catalysis or binding, as well as those involved in the ordered biochemical or physiological processes at the cell, tissue or organism level listed under biological process were screened out ($P < 0.05$). Finally, nine candidate genes were figured out in the second round of selection by KEGG pathway annotation for genes implicated in growth/development, reproduction, virulence, autophagy, proliferation, cell wall integrity, drug resistance, and immunity which are the key areas concerned by our lab”

** See Nature Portfolio's author and referees' website at www.nature.com/authors for information about policies, services and author benefits.

This email has been sent through the Springer Nature Tracking System
NY-610A-NPG&MTS

Confidentiality Statement:

This e-mail is confidential and subject to copyright. Any unauthorised use or disclosure of its contents is prohibited. If you have received this email in error please notify our Manuscript Tracking System Helpdesk team at <http://platformsupport.nature.com> .

Details of the confidentiality and pre-publicity policy may be found here <http://www.nature.com/authors/policies/confidentiality.html>

Privacy Policy | Update Profile